# Visualization of High-Dimensional Matrix Manifolds

## Abstract

Matrix manifolds play a fundamental role in machine learning, underpinning data representations (*e.g.*, linear subspaces and covariance matrices) and optimization procedures. These manifolds follow Riemannian geometry, where intrinsic geometric structure plays an important role in geometric learning algorithms. However, traditional visualization methods based on Euclidean assumptions often fail to respect such non-Euclidean structure, leading to distortions in the resulting embeddings. To address this limitation, we generalize the popular t-SNE paradigm to the context of Riemannian manifolds and apply it to three types of matrix manifolds, which are the Grassmann manifolds, Correlation manifolds, and Symmetric Positive Semi-Definite (SPSD) manifolds, respectively. By introducing Riemannian geodesics to define probability distributions between the original and target spaces, our method transforms high-dimensional manifold-valued data into low-dimensional embeddings, thereby respecting the intrinsic geometry of the data, with curvature-related properties implicitly reflected through geodesic distances, and reducing distortions caused by Euclidean approximations. This work provides a foundation for general-purpose dimensionality reduction of high-dimensional matrix manifolds. Extensive experimental comparisons with existing visualization methods across synthetic and benchmarking datasets demonstrate the efficacy of our proposal in preserving geometric properties of the data.

## 1 Introduction

Matrix-valued data, such as linear subspaces, Correlation matrices, and covariance matrices, naturally reside on non-Euclidean manifolds. While these descriptors can capture rich geometric and statistical information, their curved geometry fundamentally challenges standard visualization tools built upon Euclidean assumptions. In consequence, there is a growing need for geometry-aware visualization methods capable of faithfully preserving the intrinsic structure of manifold-valued data, thereby facilitating the development of non-Euclidean representation learning.

**Grassmann manifolds.** The Grassmann manifold $\mathrm{Gr}(d, q)$ is the set of all $q$-dimensional linear subspaces in $\mathbb{R}^d$, which can be represented by $d \times q$ orthonormal basis matrix. As a fundamental subspace descriptor, it frequently appears in algorithms such as linear regression, principal component analysis (Knudsen, 2001; Jansson & Wahlberg, 1996), low-rank matrix completion (Vidal & Favaro, 2014), and image set classification (Wang et al., 2020), image fusion (Kang et al., 2025). In addition, it demonstrates good performance in several downstream vision applications, such as action recognition (Nguyen & Yang, 2023; Chen et al., 2024), EEG decoding (Ingolfsson et al., 2020; Wang et al., 2024), facial recognition (Ingolfsson et al., 2020; Wang et al., 2021), and recommender systems (Cao et al., 2016). Nevertheless, visualizing high-dimensional Grassmannian points remains a key challenge.

**Correlation manifolds.** A Correlation matrix is a symmetric matrix that encodes linear dependencies among variables, with each entry typically representing a Pearson correlation coefficient. Its key property is scale invariance, making it suitable for settings where absolute magnitudes are irrelevant (Thanwerdas & Pennec, 2022; Thanwerdas, 2024). Taking the Electroencephalogram (EEG) analysis as an example, two electrodes may show a strong correlation despite differing signal amplitudes. Since the Correlation matrix has the ability to capture intrinsic geometric relationships independent of scale, it has been widely used in fields such as brain connectivity analysis (Varoquaux et al., 2010), finance (Rebonato & Jäckel, 2000; Marti

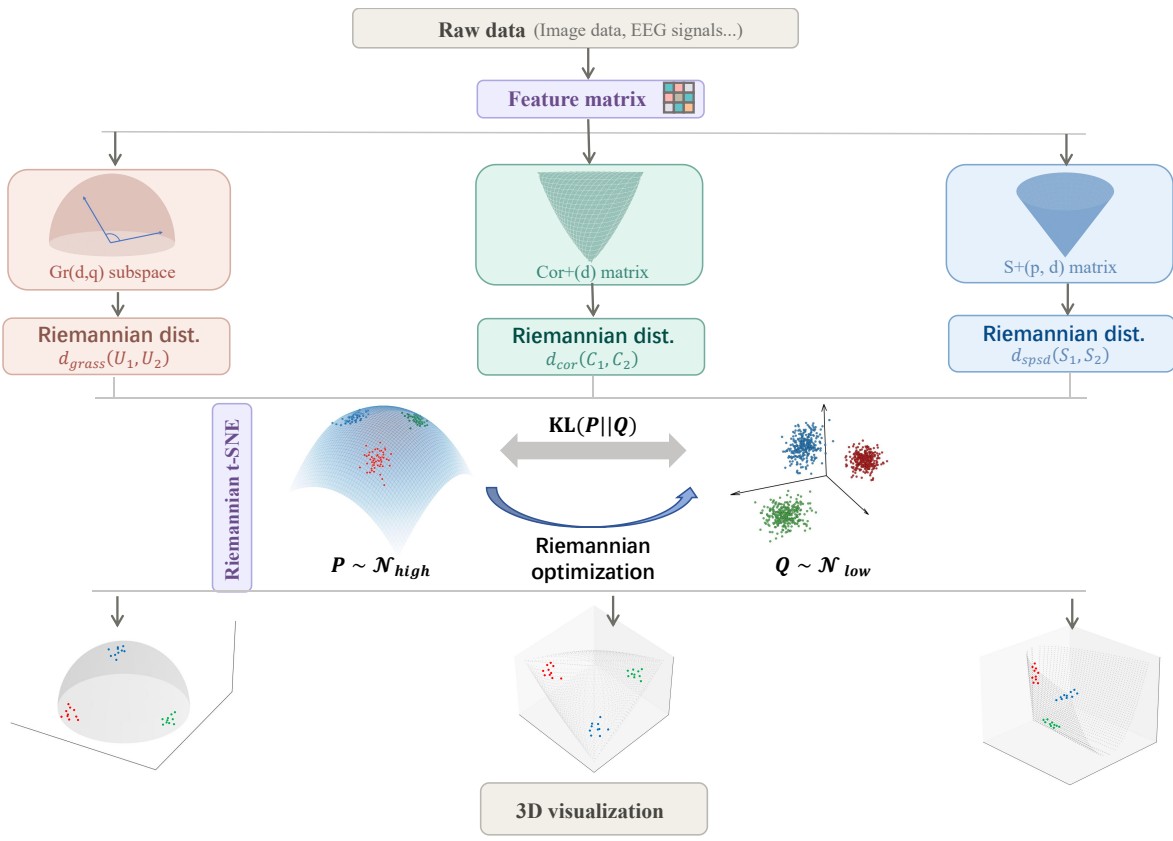

Figure 1: A visual demo of our proposed method.

et al., 2021), and Gaussian graphical models (Lauritzen, 1996; Epskamp & Fried, 2018). Currently, research on the visualization of Correlation matrices is still in its infancy.

**SPSD manifolds.** An SPSD matrix is a natural superset of SPD matrices, offering greater flexibility when dealing with low-rank or degenerate structures. SPSD matrices are prevalent in machine learning and statistics, where they commonly appear as kernel matrices and covariance matrices (Lanckriet et al., 2004; Huber & Ronchetti, 1981). They are also widely encountered in practical domains such as medical imaging (Bonnabel et al., 2013) and image set classification (Faraki et al., 2016). By encoding both geometric and statistical information, SPSD matrices provide expressive yet compact representations. This applicability makes SPSD matrices a powerful modeling tool in scenarios where strict positive definiteness of high-dimensional data cannot be guaranteed. However, this also raises the question of how to visualize the SPSD data points.

It is well known that visualization plays a central role in analyzing learned representations, evaluating model performance, and characterizing the structure of complex data. Although the aforementioned structured matrices are widely used across artificial intelligence and data science, their inherently non-Euclidean geometry poses substantial challenges for conventional analytical and visualization techniques. Classical methods like t-SNE (Van der Maaten & Hinton, 2008), while highly effective for Euclidean data, assume a flat geometric structure and therefore introduce distortions when applied to curved manifolds. In particular, a point cloud sampled from a Riemannian manifold cannot be faithfully embedded into Euclidean space without geometric distortion. This phenomenon is analogous to the well-known impossibility of projecting the Earth, a curved 2D surface, onto a flat map without altering its geometry (Mulcahy & Clarke, 2001).

To this end, several geometry-aware extensions of t-SNE have been proposed. In hyperbolic settings, CO-SNE Guo et al. (2022) generalizes t-SNE to the Poincaré disk model to account for negative curvature. Recently, GrassCaré (Li & Pimentel-Alarcón, 2024) embeds high-dimensional subspace data into a 2D Poincaré disk, enabling curvature-aware visualization of Grassmannian representations. However, GrassCaré suffers from a

fundamental curvature mismatch: the Poincaré disk has constant negative curvature, whereas the Grassmann manifold exhibits non-negative or mixed curvature. As a result, hyperbolic embeddings cannot faithfully preserve the intrinsic geometry or relational structure of the Grassmannian data.

To address these limitations, our method extends t-SNE into the Riemannian setting, enabling geometry-aware dimensionality reduction that preserves intrinsic manifold geometry. Specifically, our method produces geometry-consistent low-dimensional embeddings that respect the structural properties of the Grassmann, Correlation, and SPSD manifolds. For the low-dimensional case of the Grassmann manifold $Gr(3, 1)$, each point represents a one-dimensional subspace through the origin in $\mathbb{R}^3$, which admits a natural visualization as a unit direction vector on the Grassmann hemisphere. On the Correlation manifold, the fixed-unit diagonal and degrees of freedom concentrated in the upper triangular part allow meaningful low-dimensional embeddings with $3 \times 3$ Correlation matrices. Similarly, for SPSD matrices, we use $2 \times 2$ matrices, whose three degrees of freedom yield compact yet informative representations. Figure 1 illustrates the utilization of our method for dimensionality reduction and visualization of raw data, and the specific code implementation can be found in Figure 15 of the Appendix A.

In this paper, our main contributions are summarized as follows:

- **Manifold-specific instantiation of Riemannian t-SNE**: Building upon the established Riemannian t-SNE paradigm, we instantiate a geometry-aware visualization framework on three practically important matrix manifolds: the Grassmann, Correlation, and low-rank SPSD manifolds. The key contribution lies in adapting this paradigm to distinct geometric settings by designing appropriate manifold-specific Riemannian operators.

- **Geometry-aware visualization on the Grassmann manifold**: For subspace-valued data, we incorporate multiple Grassmannian metrics to investigate how different geometric distances influence the resulting visualizations. This provides a more geometry-consistent alternative to Euclidean visualization methods that may suffer from curvature mismatch.

- **Visualization on the underexplored Correlation and SPSD manifolds**: For Correlation and fixed-rank SPSD matrices, we develop manifold-aware visualization strategies that respect their intrinsic constraints, including a unit-diagonal correlation structure and a low-rank positive-semidefinite geometry. Experiments on these two specific matrix representations show that our method has better applicability.

- **Experimental evaluation and computational analysis**: We evaluate the proposed framework on synthetic datasets, image sets, EEG signals, visual benchmarks, and network features. In addition, we provide both theoretical complexity analysis and empirical runtime measurements across all three manifold settings to verify the scalability and practical applicability of the proposed method.

## 2 Related Work

Dimensionality Reduction (DR) is a core task in machine learning, aiming to reveal low-dimensional structures in high-dimensional data. DR methods are broadly categorized as linear, nonlinear, or Riemannian.

**Linear DR methods.** These methods project data onto subspaces that preserve key variance or discriminative information. Principal Component Analysis (PCA) (Hastie et al., 2009) identifies directions of maximum variance, while Independent Component Analysis (ICA) (Hastie et al., 2009) extracts statistically independent components. Additionally, Linear Discriminant Analysis (LDA) (Izenman, 2013), a supervised approach, seeks projections that maximize class separability.

**Nonlinear DR methods.** These approaches assume that data lies on a nonlinear manifold and aim to preserve its intrinsic geometry in the embedding. Representative methods include Multidimensional Scaling (MDS) (Borg & Groenen, 2007), Kernel PCA (Schölkopf et al., 1997), Locally Linear Embedding (LLE) (Rowes, 2000), Isometric Feature Mapping (ISOMAP) (Tenenbaum et al., 2000), Diffusion Maps (DMaps) (Coifman & Lafon, 2006), and Uniform Manifold Approximation and Projection (UMAP) (McInnes et al., 2018). These methods seek to preserve the neighborhood or topological relationships during the embedding.

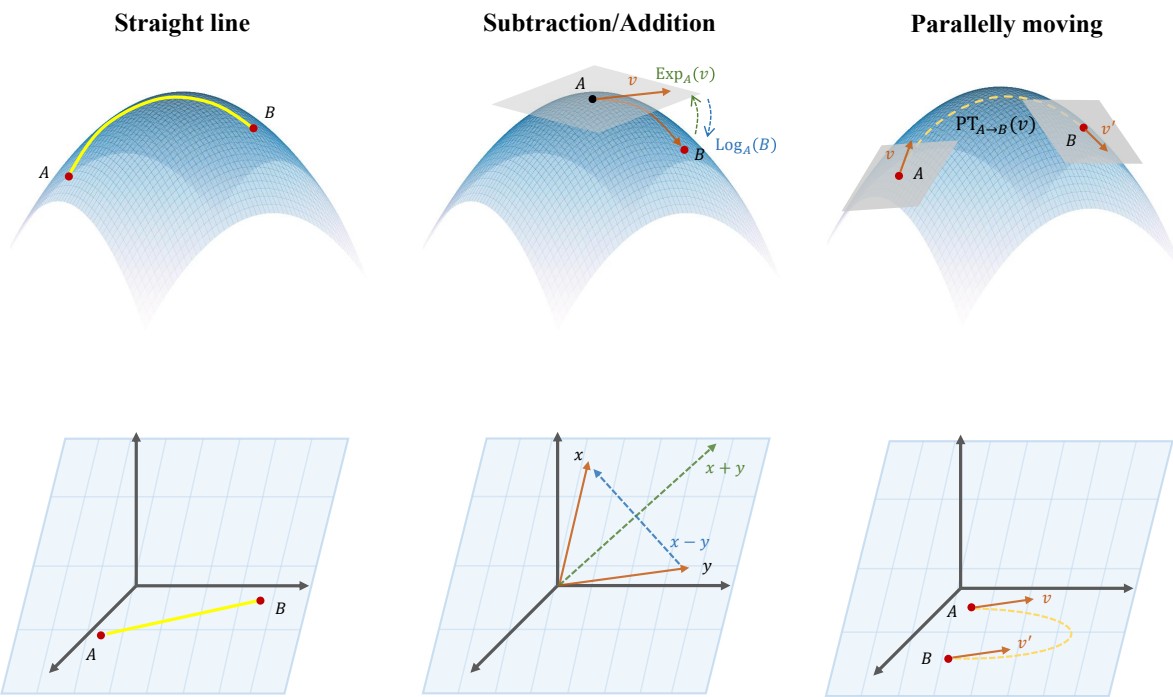

Figure 2: Illustration of the differences between the basic operations defined in Euclidean space (shown in the second row) and those on Riemannian manifolds (shown in the first row).

**Riemannian DR methods.** Early explorations of Riemannian geometry in information visualization laid the foundation for manifold-valued DR. For instance, Brand (2002) proposed a method by decomposing high-dimensional data into locally linear patches and explicitly merging them to preserve local curvature. Around the same time, Peltonen et al. (2005) introduced an information-geometric approach that estimates Riemannian metrics from Fisher information matrices to improve topology preservation in exploratory analysis tools such as SOM and Sammon's mapping. Following these foundational works, several methods generalize DR to curved manifolds. For example, Principal Geodesic Analysis (PGA) (Fletcher et al., 2004) extends PCA by modeling variance along geodesics. Hessian LLE (Goh & Vidal, 2008) has also been applied to curved spaces. In hyperbolic space, HoroPCA (Chami et al., 2021) captures hierarchical structure by identifying informative directions in negatively curved spaces, while CO-SNE (Guo et al., 2022) develops t-SNE to the Poincaré disk for visualizing hierarchical data. For SPD matrices, de Surrel et al. (2025) extends t-SNE and MDS by introducing Riemannian geometry, such as the Affine-Invariant Riemannian Metric (AIRM) and the Log-Euclidean Metric (LEM) associated with the SPD manifolds. On the Grassmann manifolds, GrassCaré (Li & Pimentel-Alarcón, 2024) embeds subspace data into a 2D Poincaré disk, while GDMaps (Dos Santos et al., 2022) generalizes DMaps to the Grassmannian setting by defining appropriate distance measures and projection kernels.

However, these methods typically map curved manifold-valued data into Euclidean or fixed non-Euclidean spaces, without explicitly enforcing the preservation of the source manifold's intrinsic Riemannian structure. Consequently, geometric properties specific to different curved manifolds may not be fully retained in the resulting embeddings. Motivated by these limitations, the goal of this paper is to develop a geometry-aware visualization framework for curved manifolds, enabling geometry-preserved manifold-to-manifold embedding mapping.

## 3 Preliminary

In this section, we will briefly introduce Riemannian geometry and the three matrix manifolds involved.

Table 1: Comparison of basic operations (Chen et al., 2025) defined in Euclidean space and Riemannian manifolds.

| Operation | Euclidean space | Riemannian manifolds |
|---|---|---|
| Straight line | Straight line | Geodesic |
| Subtraction | $\overrightarrow{xy} = y - x$ | $\overrightarrow{xy} = \text{Log}_x(y)$ |
| Addition | $y = x + \overrightarrow{xy}$ | $y = \text{Exp}_x(\overrightarrow{xy})$ |
| Parallelly moving | $v \to v$ | $\text{PT}_{x \to y}(v)$ |

**Riemannian geometry.** Riemannian geometry provides a principled framework for analyzing data that reside on curved, non-Euclidean spaces, known as manifolds. To build intuition, it is helpful to contrast it with standard Euclidean geometry. In Euclidean space, the shortest path between two points is a straight line, and vector operations are globally defined. For example, subtraction can be written as $y - x$, while addition is expressed as $x + \overrightarrow{xy}$. On a manifold, however, these notions must be reformulated to account for the underlying curvature. In particular, straight lines are replaced by *geodesics*, i.e., the shortest paths on the manifold. A classic example is the Earth's surface, where the shortest path between two locations is a great-circle arc rather than a straight line passing through the Earth's interior. At the same time, the surface can be locally approximated by a flat Euclidean space, known as the tangent space of each point on the manifold. Based on this local linear structure, the analogue of vector subtraction is given by the *logarithmic map* $\text{Log}_x(y)$, which maps a point $y$ to the tangent space at $x$. Conversely, the analogue of vector addition is realized by the *exponential map* $\text{Exp}_x(v)$, which maps a tangent vector $v$ back onto the manifold. Since tangent spaces are point-dependent, vectors at different locations cannot be compared directly and must instead be related via *parallel transport*. Tab. 1 compares the basic operations defined in Euclidean space and Riemannian manifolds, while Fig. 2 provides an intuitive illustration of these differences for better understanding.

**Grassmann Manifolds.** The Grassmann manifold $\text{Gr}(d, q)$ is the set of all $q$-dimensional subspaces of $\mathbb{R}^d$. It is a smooth, compact manifold of dimension $q(d - q)$ (Bendokat et al., 2024). Each subspace on $\text{Gr}(d, q)$ can be represented by a matrix $U \in \mathbb{R}^{d \times q}$ with orthonormal columns, *i.e.*, $U^\top U = I_q$, where $I_q$ is the $q \times q$ identity matrix. Different bases of the same subspace are associated by an orthogonal matrix, forming an equivalence class shown below:

$$[U] = \left\{ \tilde{U} \mid \tilde{U} = UO, \, O \in \mathcal{O}(q) \right\}. \tag{1}$$

We refer to this as the orthonormal basis (ONB) representation and use $U$ and $[U]$ interchangeably by abuse of notation.

**Correlation Manifolds.** A Correlation matrix is obtained by normalizing a covariance matrix by its variances (Thanwerdas & Pennec, 2022). Given covariance matrix $\Sigma = (\text{Cov}(X_i, X_j))_{1 \le i,j \le d}$, the corresponding Correlation matrix has entries

$$C_{ij} = \frac{\Sigma_{ij}}{\sqrt{\Sigma_{ii}} \sqrt{\Sigma_{jj}}} = (\Upsilon^{-1/2})_{ii} \Sigma_{ij} (\Upsilon^{-1/2})_{jj}, \tag{2}$$

where $\Upsilon = \text{Diag}(\Sigma)$. This defines a mapping from SPD matrices to full-rank Correlation matrices: $\Sigma \in \text{Sym}^+(d) \longmapsto \Upsilon^{-1/2} \Sigma \Upsilon^{-1/2} \in \text{Cor}^+(d)$.

**SPSD Manifolds.** The SPSD manifold $\mathcal{S}_+(p, d)$ is the set of SPSD matrices of size $d \times d$ and rank $p \le d$:

$$\mathcal{S}_+(p, d) = \left\{ S \in \mathbb{R}^{d \times d} \mid S = S^\top, \, S \succeq 0, \, \text{rank}(S) = p \right\}. \tag{3}$$

When $p = d$, it reduces to the SPD manifold. This space admits a quotient structure: $\mathcal{S}_+(p, d) \cong \mathbb{R}_*^{d \times p} / \mathcal{O}_p$, where $\mathbb{R}_*^{d \times p}$ denotes full-rank matrices. Given any $F \in \mathbb{R}_*^{d \times p}$, the mapping $\Phi : F\mathcal{O}_p \mapsto FF^\top$ is well-defined and induces a smooth bijection between the quotient space and $\mathcal{S}_+(p, d)$ (Massart & Absil, 2020).

# 4 Riemannian Dimensionality Reduction

t-SNE is a widely used method for nonlinear dimensionality reduction in Euclidean space. It transforms pairwise distances into conditional probabilities that reflect pointwise similarities. Given a high-dimensional point $x_i$, the probability of choosing $x_j$ as its neighbor is computed by:

$$p_{j|i} = \frac{\exp\left(-\|x_i - x_j\|^2/2\sigma_i^2\right)}{\sum_{k \neq i} \exp\left(-\|x_i - x_k\|^2/2\sigma_i^2\right)}, \tag{4}$$

where $\sigma_i$ is the Gaussian variance centered at $x_i$. The symmetric joint probability is then defined as $p_{ij} = \frac{p_{i|j}+p_{j|i}}{2N}$, where $N$ signifies the number of data points.

To mitigate the "crowding problem" inherent in previous visualization methods such as SNE (Hinton & Roweis, 2002), t-SNE introduces a heavy-tailed t-distribution in the low-dimensional data space. This modification enhances repulsion between dissimilar points and prevents overcrowding in the embedding process. The low-dimensional similarity between $y_i$ and $y_j$ in SNE is quantified as:

$$q_{ij} = \frac{\exp\left(-\|y_i - y_j\|^2\right)}{\sum_{k \neq i} \exp\left(-\|y_i - y_k\|^2\right)}, \tag{5}$$

where $y_i$ denotes the low-dimensional representation of $x_i$.

Finally, t-SNE minimizes the Kullback-Leibler (KL) divergence between the high-dimensional joint distribution and the low-dimensional distribution:

Considering that t-SNE is unable to be directly used in non-Euclidean space, we build upon the Riemannian t-SNE paradigm established in prior work (de Surrel et al., 2025; Guo et al., 2022)and instantiate it for three geometrically distinct matrix manifolds. Specifically, we similarly constructed high-dimensional and low-dimensional distributions and optimized them using the KL divergence. Our primary contributions lie in the manifold-specific geometric developments required to realize this framework on the Grassmann, Correlation, and SPSD manifolds, each of which demands a tailored geodesic distance, a corresponding Riemannian gradient derivation, and an appropriate retraction operation that keeps embeddings on the target manifold. Specifically, our visualization framework comprises four key components:

(1) **manifold-aware data modeling**, which embeds the input data onto their corresponding Riemannian manifolds (*e.g.*, Grassmann, Correlation, and SPSD manifolds); (2) **intrinsic similarity measurement**, where Riemannian distances are employed to faithfully quantify pairwise relationships; (3) **data distribution construction** on both the high-dimensional and low-dimensional manifolds; and (4) **geometry-aware optimization** for low-dimensional embedding. The following sections elaborate on each component and describe how they are integrated to form our proposed Riemannian t-SNE framework.

## 4.1 Manifold-aware Data Modeling

To leverage our proposed method, the data need to be transformed onto their corresponding matrix manifolds, firstly. A key observation is that many matrix manifolds used for modeling time-series data or image sets naturally arise from second-order statistics. In particular, the covariance representation provides a unified, canonical, and widely adopted way to capture the spatiotemporal dependencies among different data regions, serving as a common bridge for constructing Grassmannian, Correlation, and SPSD elements (Wang et al., 2024; Hu et al., 2025; Wang et al., 2025). For clarity, we illustrate this modeling process using EEG signals as a typical example. Let $X \in \mathbb{R}^{d \times T}$ be an EEG trial, in which $d$ and $T$ signify the number of channels (e.g., electrodes) and the length of time dimension per trial, respectively. Then, the corresponding sample covariance matrix can be computed by $\Sigma = \frac{1}{T-1}\bar{X}\bar{X}^\top$, where $\bar{X}$ denotes the matrix obtained by centering $X$ along the time dimension.

Accordingly, different manifold embeddings can be utilized to obtain the corresponding manifold-valued data:

- **Grassmann Manifold:** The column space spanned by the dominant $q$ eigenvectors of $\Sigma$ defines a point on $\mathrm{Gr}(d, q)$. This is realized by Eigenvalue Decomposition, *i.e.*, $\Sigma = U\Lambda U^\top$, where $\Lambda =$

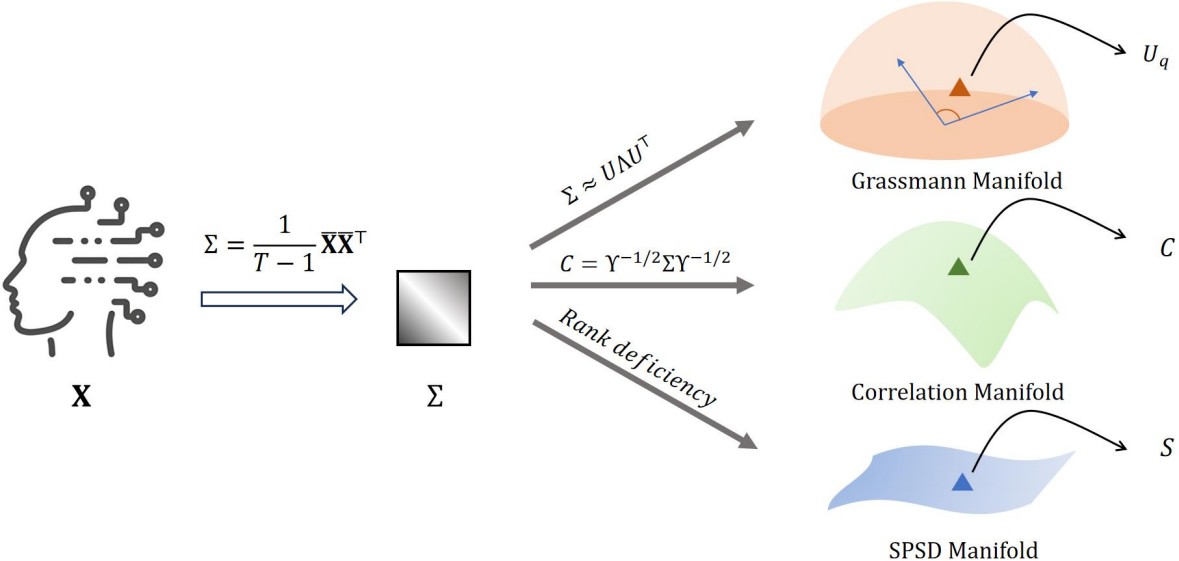

Figure 3: An example of modeling EEG data onto different matrix manifolds. For simplicity, we omit some of the preprocessing steps required for EEG data in this figure.

$\operatorname{diag}(\sigma_1, \ldots, \sigma_d)$ with $\sigma_1 \geq \cdots \geq \sigma_d \geq 0$ and $U = [u_1, \ldots, u_d] \in \mathbb{R}^{d \times d}$ signify the eigenvalue and the corresponding eigenvector matrices, respectively. The truncated matrix $U_q = [u_1, \ldots, u_q] \in \mathbb{R}^{d \times q}$ spans a $q$-dimensional linear subspace, represented as $[U_q] \in \operatorname{Gr}(d, q)$.

- **Correlation Manifold:** Normalizing $\Sigma$ by its diagonal yields $C = \Upsilon^{-1/2} \Sigma \Upsilon^{-1/2}$, where $\Upsilon = \operatorname{Diag}(\Sigma)$ is a diagonal matrix consisting of the main diagonal elements of $\Sigma$. At this time, $C$ resides on the Correlation manifold, a submanifold of SPD matrices with unit diagonal entries.

- **SPSD Manifold:** The covariance matrix $\Sigma$ is inherently SPSD, encoding either full-rank or low-rank structure depending on the relationship between $d$ and $T$. Its rank satisfies $\operatorname{rank}(\Sigma) \leq \min(d, T-1)$. When $T - 1 \geq d$, $\Sigma$ is typically full-rank and lies in the interior of the SPSD cone $\mathcal{S}_+^d$, *i.e.*, the SPD manifold $\mathcal{S}_{++}^d$. When $T - 1 < d$, $\Sigma$ is rank-deficient and resides on the boundary of $\mathcal{S}_+^d$. In the latter case, $\Sigma$ lies on a fixed-rank SPSD manifold.

For an intuitive understanding, please refer to the modeling process in Fig. 3.

### 4.2 Intrinsic Similarity Measurement

Unlike Euclidean spaces, where distances are computed along straight lines, Riemannian manifolds require intrinsic geodesic distances that reflect their underlying curvature. As a consequence, Euclidean metrics cannot be used in the proposed Riemannian t-SNE framework, as they disregard manifold geometry and will produce distorted similarity measurements (see Fig. 4). To ensure that pairwise relationships are faithfully reflected, we employ intrinsic Riemannian geodesic distances as the basis for similarity computation.

For the Grassmann manifold, the geodesic distance between two subspaces $U_1 \in \mathbb{R}^{d \times q}$ and $U_2 \in \mathbb{R}^{d \times q}$ is defined as (Ye & Lim, 2016):

$$d_{grass}(U_1, U_2) = \left( \sum_{i=1}^{q} \theta_i^2 \right)^{1/2}, \tag{6}$$

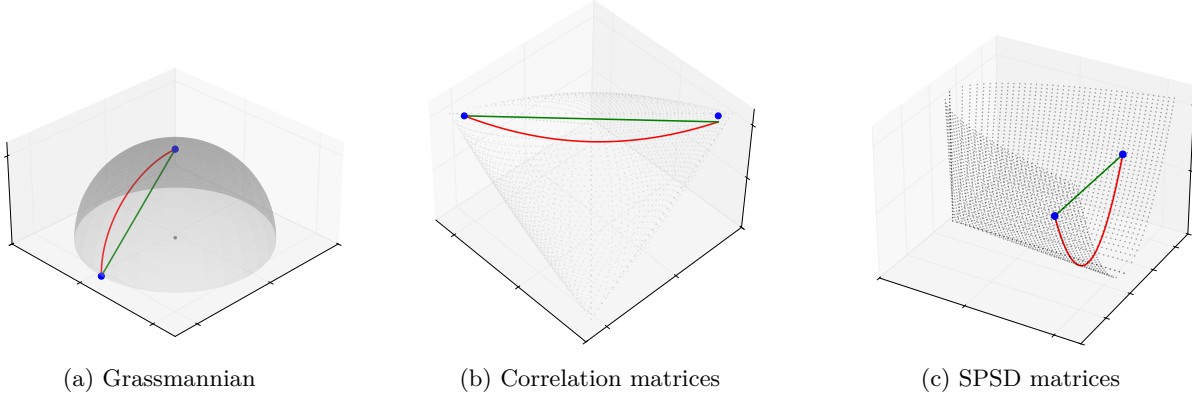

(a) Grassmannian       (b) Correlation matrices       (c) SPSD matrices

Figure 4: Comparison of geodesic and Euclidean distances on different manifolds. (a) Different distances of a Grassmann manifold. (b) Different distances of a Correlation manifold. (c) Different distances of an SPSD manifold. Wherein, the red curve denotes the intrinsic geodesic, while the green line indicates the ambient Euclidean straight line.

where $\theta_i$ $(i = 1 \rightarrow q)$ are the principal angles between two subspaces. These angles are obtained from the singular value decomposition (SVD) of $U_1^\top U_2$. Let $\sigma_i$ denote the $i$-th singular value, the corresponding principal angle is then given by $\theta_i = \arccos(\sigma_i)$. This distance captures the minimal rotational discrepancy between subspaces. As it is computed from the principal angles, it respects the underlying Riemannian geometry of the Grassmann manifold and provides a natural measure of subspace similarity.

For the Correlation manifolds, we adopt the Poly-Hyperbolic-Cholesky (PHC) metric (Thanwerdas & Pennec, 2022) to characterize the similarity between any two full-rank Correlation matrices:

$$d_{cor}(C_1, C_2) = \left( \sum_{i=2}^n \left[ \arccos \left( -Q \left( L_{1i}^\top, {L'_{2i}}^\top \right) \right) \right]^2 \right)^{1/2}, \tag{7}$$

where $L_1 = \text{Chol}(C_1)$ and $L_2 = \text{Chol}(C_2)$ denote the Cholesky factors of $C_1$ and $C_2$, respectively. Here, $L_{1i}$ and $L_{2i}$ represent the $i$-th rows of the lower triangular matrices, and $Q(x, y) = \sum_{j=1}^k x_j y_j - x_{k+1} y_{k+1}$ signifies the Lorentzian inner product.

The PHC metric is derived from the geodesic structure of hyperbolic geometry. To be specific, each Correlation matrix is mapped (via its Cholesky decomposition) into a product space of hyperbolic manifolds $\mathbb{H}^{i-1}$, where $L_{1i}^\top$ and $L_{2i}^\top \in \mathbb{R}^i$ are interpreted as points in hyperbolic space. Geodesic distances are computed in this hyperbolic product space and subsequently pulled back to the Correlation manifold via a smooth diffeomorphism, yielding an intrinsic and geometry-consistent similarity measure.

For the SPSD manifolds, we follow Massart & Absil (2020) and exploit the quotient-geometric distance for similarity measurement:

$$d_{spsd}(S_1, S_2) = [\text{tr}(S_1) + \text{tr}(S_2) - 2\,\text{tr}(S^*)]^{1/2}, \tag{8}$$

where $S^* = \left( S_1^{1/2} S_2 S_1^{1/2} \right)^{1/2}$ and $\text{tr}(\cdot)$ represents the matrix trace. This metric arises from the quotient geometry of the SPSD matrices, obtained by projecting the horizontal components of the equivalence-class representatives onto the quotient space. Importantly, the resulting geodesic distance preserves their intrinsic low-rank structure. In other words, the geodesic curve connecting $S_1$ and $S_2$ remains within the same fixed-rank stratum, ensuring that the manifold geometry does not artificially increase the rank or alter the underlying subspace structure.

In addition, several alternative Riemannian metrics are available on both the Grassmann and Correlation manifolds. Detailed descriptions of them are provided in Appendix A.1 and Appendix A.2. The metrics

introduced above are the primary choices in the proposed framework. In the experiments, we further compare multiple metrics to assess their influence and to validate the rationale behind our selected benchmark.

### 4.3 Data Distribution Construction.

A core component of the proposed Riemannian t-SNE framework is the construction of probability distributions that characterize pairwise similarities on curved manifolds. In Euclidean t-SNE, these probabilities are derived from Gaussian and Student t-distributions. However, they are defined with respect to Euclidean distance and therefore cannot be directly applied to manifold-valued data. To tackle this limitation, we replace Euclidean distances with intrinsic geodesic distances and generalize the corresponding probability distributions accordingly. This construction follows the paradigm of Guo et al. (2022) and de Surrel et al. (2025), generalized here to the Grassmann, Correlation, and SPSD manifolds through their respective geodesic distances defined in Section 4.2.

We first introduce the Riemannian normal distribution, a natural analogue of the Euclidean Gaussian that maximizes entropy subject to prescribed expectation and variance (Pennec, 2006). Given a Fréchet mean $\mu$ and dispersion parameter $\sigma$, the distribution takes the following form

$$\mathcal{N}(X|\mu, \sigma^2) = \frac{1}{Z} \exp\left(-\frac{\delta(X, \mu)^2}{2\sigma^2}\right), \tag{9}$$

where $\delta(X, \mu)$ denotes the geodesic distance on the manifold and $Z$ is the normalization constant. This construction ensures that probability mass is assigned according to the intrinsic geometry of the manifold, rather than a Euclidean approximation.

To define similarities in the low-dimensional manifold, we extend the Student t-distribution with one degree of freedom using the same geodesic principle. The resulting distribution is given by

$$q_{ij} = \frac{(1 + \delta(Y_i, Y_j)^2)^{-1}}{\sum_{k \neq l}(1 + \delta(Y_k, Y_l)^2)^{-1}}, \tag{10}$$

where $Y_i$ represents the learned low-dimensional embedding of the $i$-th data point. This heavy-tailed distribution preserves the desirable separation properties of the original t-SNE, while respecting the manifold geometry through $\delta(\cdot, \cdot)$.

Finally, to align similarities between the high-dimensional and low-dimensional manifolds, we minimize the KL divergence between the high-dimensional distribution $P$ and the low-dimensional distribution $Q$.

$$\mathcal{L} = \sum_i \text{KL}(P \parallel Q) = \sum_i \sum_j p_{ij} \log \frac{p_{ij}}{q_{ij}}. \tag{11}$$

This objective encourages points that are close in the original manifold to remain close after embedding, while allowing distant points to be placed more flexibly. In other words, it retains the key behaviors of Euclidean t-SNE, but reformulated in a manner that is consistent with Riemannian geometry.

### 4.4 Geometry-aware Optimization.

To obtain the low-dimensional manifold-valued embeddings, we minimize the KL divergence loss $\mathcal{L}$ through Riemannian Stochastic Gradient Descent (RSGD). The general RSGD framework on smooth manifolds is well-established in Absil et al. (2008) and Boumal (2023), where each iteration typically consists of computing the Riemannian gradient and applying a retraction-based update to keep the iterates residing on the underlying manifold. Building on these established geometric tools, we instantiate the Riemannian t-SNE objective on matrix manifolds, enabling geometry-consistent dimensionality reduction and visualization of high-dimensional matrix manifolds.

The gradient of $\mathcal{L}$ with respect to an embedding $Y_i$ is given by

$$\nabla_{Y_i}\mathcal{L} = 2\sum_j \frac{\partial \mathcal{L}}{\partial \delta_{ij}} \nabla_{Y_i}\delta_{ij} = 4\sum_j \delta_{ij}(p_{ij} - q_{ij})(1 + \delta_{ij}^2)^{-1}\nabla_{Y_i}\delta_{ij}, \tag{12}$$

where $\delta_{ij}$ denotes the Riemannian distance between $Y_i$ and $Y_j$, and $\nabla_{Y_i}\delta_{ij}$ is the Riemannian gradient of the distance function.

To illustrate the computation of $\nabla_{Y_i}\delta_{ij}$, we consider the Grassmann manifold case using the geodesic distance in Eq. 6. For two subspaces $Y_i$ and $Y_j$, let $\{\sigma_l\}$ and $\{u_l, v_l\}$ denote the $l$-th singular values and singular vectors of $Y_i^\top Y_j$. Following the gradient computation on the Grassmann manifold established in Edelman et al. (1998), the Riemannian gradient can be obtained by projecting the Euclidean gradient onto the tangent space $T_{Y_i}\mathrm{Gr}(d, q)$ at $Y_i$:

$$\nabla_{Y_i}\delta_{ij} = (I - Y_iY_i^\top)\left(\sum_{l=1}^q \frac{-\theta_l}{\sqrt{1 - \sigma_l^2}}u_lv_l^\top Y_j^\top\right). \tag{13}$$

where $\theta_l = \arccos(\sigma_l)$ are the principal angles, and $(I - Y_iY_i^\top)$ signifies the orthogonal projection onto the tangent space $T_{Y_i}\mathrm{Gr}(d, q)$.

Once the Riemannian gradient is computed, the parameters can be updated along the geodesic on the Grassmann manifold under the *Retraction* operation. This can be expressed as:

$$R(Y_i - \eta\nabla_{Y_i}\mathcal{L}) = \mathrm{Exp}_{Y_i}(-\eta\nabla_{Y_i}\mathcal{L}), \tag{14}$$

where $\eta$ is the learning rate and $\mathrm{Exp}(\cdot)$ signifies the exponential map. Given a tangent vector $\Delta$, the exponential map is formulated as:

$$\mathrm{Exp}_{Y_i}(\Delta) = (Y_iV\cos\Lambda + U\sin\Lambda)V^\top, \tag{15}$$

where $U\Lambda V^\top$ is the SVD of $\Delta$. This update ensures that the embedding remains on the Grassmann manifold throughout optimization. Additional implementation details are provided in Appendix A.3.

### 4.5 Link to Related Works

The proposed Riemannian t-SNE framework shares a common paradigm with two closely related works, CO-SNE (Guo et al., 2022) and SPD-SNE (de Surrel et al., 2025). Specifically, all three methods replace Euclidean distances with intrinsic geodesic distances to construct geometry-aware probability distributions and minimize a KL divergence between high-dimensional and low-dimensional similarity distributions.

However, these methods differ substantially in the target manifolds they address and the geometric machinery they employ. Specifically, CO-SNE embeds high-dimensional data into the Poincaré disk, a model of hyperbolic space with constant negative curvature. Although this is well-suited for hierarchical data, it is geometrically mismatched with Grassmann manifolds, which exhibit non-negative or mixed sectional curvature. SPD-SNE, on the other hand, extends t-SNE to the manifold of full-rank SPD matrices, leveraging metrics such as the AIRM and the LEM, both of which are specifically tailored to the SPD setting. It can be seen that neither method can be directly applied to the three matrix manifolds involved in this work.

As a countermeasure, our method extends the established manifold-valued visualization paradigm to three geometrically distinct and practically important matrix manifolds. For the Grassmann manifold, we employ the geodesic distance derived from principal angles. For the Correlation manifold, we adopt the PHC metric, whose geodesic structure arises from a diffeomorphic mapping to a product of hyperbolic spaces, thereby providing a geometry that is both intrinsically consistent and computationally tractable. For the fixed-rank SPSD manifold, we exploit a quotient-geometric distance that preserves the rank constraint along geodesics. For each matrix manifold, specific Riemannian gradients and retraction operations need to be computed and exploited to ensure that the learned embeddings remain on the target manifold throughout the optimization process.

# 5 Experiments and Results

For each manifold, three types of experiments are conducted to verify the effectiveness of our method: 1) synthetic datasets sampled from a normal distribution, 2)benchmarking datasets, as well as intermediate representations (network features) extracted from designated layers, and 3) trustworthiness-based evaluation to quantify the extent to which global and local structures are preserved during dimensionality reduction.

## 5.1 Experiments on the Grassmann manifold

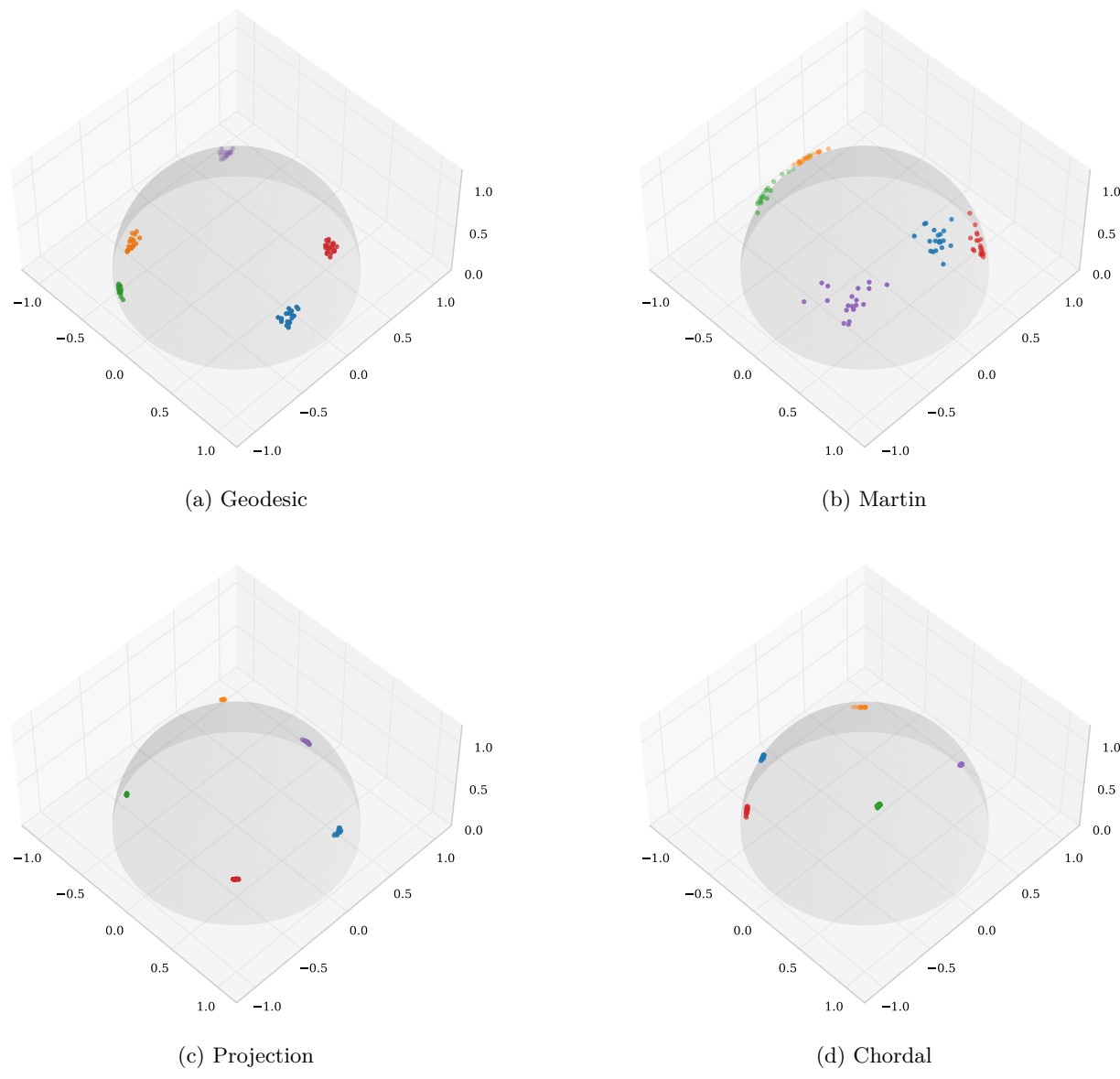

(a) Geodesic

(b) Martin

(c) Projection

(d) Chordal

Figure 5: Visualization results of synthesized data points at different Grassmannian distances.

**Setup.** We compare our method against three baselines: GDMaps (Dos Santos et al., 2022), UMAP (McInnes et al., 2018), and t-SNE (Van der Maaten & Hinton, 2008). Both t-SNE and UMAP are widely used nonlinear dimensionality reduction methods that rely on Euclidean distances to preserve data distributions in a low-dimensional space. GDMaps extends diffusion maps to the Grassmann manifold by incorporating manifold-specific distances and kernels. We adopt the implementations of t-SNE and UMAP in Pedregosa

et al. (2011) with default parameters. For GDMaps, we take the implementation from Li & Pimentel-Alarcón (2024). Perplexity is an important parameter in t-SNE. In our method, we set it to $\frac{3}{4}N$, where $N$ denotes the number of data points, consistent with the settings in de Surrel et al. (2025).

**Synthetic experiments.** To make an intuitive comparison between different distances on the Grassmann manifold (see Appendix A.1 for their definitions), we visualize synthesized data points using the proposed method under varying distance choices. Specifically, we simulate 5 point clusters in a 10-dimensional space. Each cluster contains 20 samples drawn from a Riemannian normal distribution on the Grassmann manifold $Gr(7, 2)$, where each point is represented as a $7 \times 2$ column-orthonormal matrix. The cluster centers are first sampled from a Riemannian normal distribution with unit variance. Around each center, 20 samples are generated by adding small Gaussian noise (variance 0.1), followed by orthogonalization to ensure that these points reside on the Grassmann manifold. The results are shown in Fig. 5.

It can be observed that different distances lead to distinct embedding patterns across clusters. Specifically, the results in Fig. 5c and Fig. 5d lead to a severe collapse of high-dimensional data points belonging to the same category into nearly single points in the low-dimensional space. This phenomenon suggests that the projection and chordal distances fail to preserve the intra-cluster geometric structure, which is unfavorable for faithful manifold visualization. In contrast, Fig. 5b excessively stretches the intra-cluster structure, resulting in an over-dispersed embedding that blurs the boundaries between different categories and hinders discriminability. As expected, Fig. 5a provides a more balanced and structure-preserving visualization. This is because the geodesic distance corresponds to the length of geodesic curves induced by the canonical Riemannian metric on the Grassmann manifold. Unlike the chordal and projection distances, which rely on extrinsic Euclidean embeddings, the geodesic distance is defined intrinsically on the manifold itself. Consequently, it more faithfully captures the underlying curvature and geometric relationships of Grassmannian data points. Based on these observations, the geodesic distance is adopted in all subsequent experiments.

**Benchmarking datasets.** We further evaluate the proposed method on the MNIST dataset by visualizing the image groups represented as linear subspaces. Following the preprocessing procedures in Li & Pimentel-Alarcón (2024), we randomly sample 60 images from each digit class in the training set and form multiple groups. Each group is processed via Singular Value Decomposition, and the top 3 singular vectors are retained to construct a rank-3 subspace representation. Given the original image size of $28 \times 28$, each group is thus modeled as a point on the Grassmann manifold $Gr(784, 3)$. We sample 50 such subspaces per class.

For ease of observation, the learned low-dimensional embeddings are displayed from a top-down view of the Grassmann hemisphere. As shown in Fig. 6, both our method and GDMaps produce well-separated clusters corresponding to different digit classes, whereas UMAP and t-SNE fail to yield clear class separation. This is mainly because UMAP and t-SNE operate in a flat Euclidean embedding space, which is not well suited for preserving the intrinsic Riemannian geometry of Grassmannian data, often leading to geometric distortions during dimensionality reduction.

Although GDMaps preserves global manifold structure by emphasizing diffusion distances, it exhibits limited discriminability when separating locally similar clusters, resulting in partial overlap among certain digit classes. This limitation is mainly attributed to the subspace construction stage rather than the diffusion process itself. Specifically, GDMaps relies on a truncated $q$-dimensional subspace representation for each sample prior to diffusion (Dos Santos et al., 2022). When $q$ is small, geometrically informative directions may be irreversibly discarded, leading to a distorted Grassmannian geometry and consequently degenerate diffusion embeddings. In contrast, our method avoids hard spectral truncation and preserves local neighborhood relationships in a geometrically aligned manner, yielding embeddings that are less sensitive to the choice of subspace dimensionality and more robust for visualization.

Overall, these results demonstrate the effectiveness of our method in producing compact and well-separated embeddings, particularly for visually similar digit classes.

## 5.2 Experiments on the Correlation manifold

**Setup.** For experiments on the Correlation manifolds, we adopt a similar set of baselines as those used in the Grassmannian setting, including t-SNE and UMAP. Additionally, MDS (Borg & Groenen, 2007) is

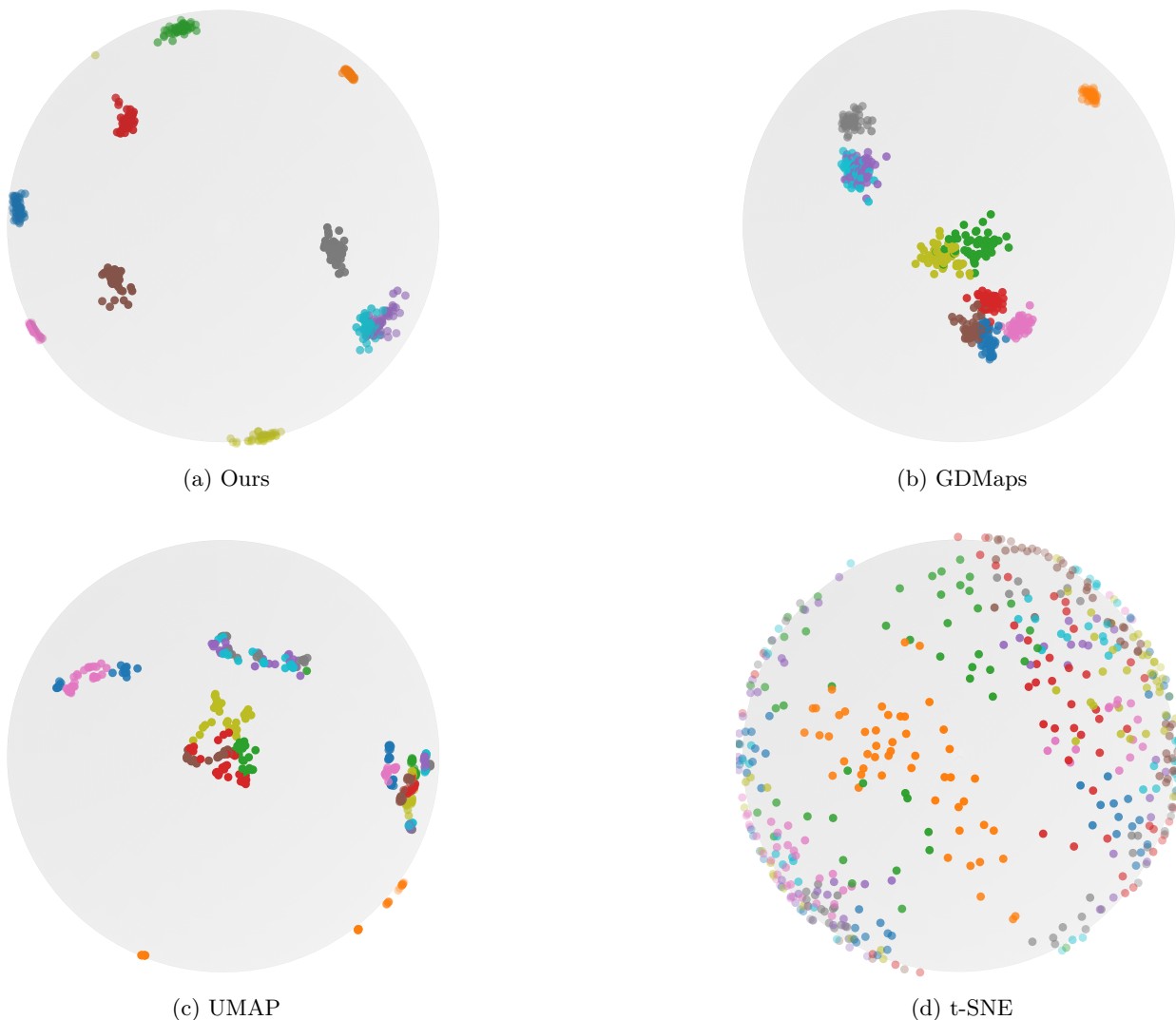

Figure 6: Visualization results of different methods on subspaces of the MNIST dataset.

incorporated as a classical baseline. The implementations and hyperparameter settings of all the involved comparative methods are kept consistent with those used in the previous experiments.

**Synthetic experiments.** Following the experimental protocol on the Grassmann manifold, we evaluate the impact of different Riemannian distances on the visualization of full-rank Correlation matrices using a synthetic dataset. It consists of three clusters, each containing 50 samples represented as $5 \times 5$ Correlation matrices, which reside on a 10-dimensional Correlation manifold. Specifically, these matrices are generated by sampling from a Riemannian normal distribution with variance 0.2, followed by normalizing the diagonal entries to enforce unit variance.

The visualization results are illustrated in Fig. 7. It can be observed that the PHC distance yields clearer separation among different clusters compared to other Riemannian distances. We argue that it is mainly attributed to the favorable geometric and computational properties of the PHC metric on the Correlation manifold. In particular, the PHC distance defines a true geodesic metric and induces a non-positively curved Riemannian structure. Moreover, it admits unique Riemannian means and well-defined logarithmic mappings, while remaining numerically stable and computationally efficient (Thanwerdas & Pennec, 2022; Thanwerdas, 2024).

Based on these observations, we designate PHC as the benchmark distance on the Correlation manifold in all subsequent experiments.

**Features from a manifold-valued EEG Network.** We further evaluate the proposed method on the MAMEM-SSVEP-II dataset (Georgiadis et al., 2016), which contains EEG recordings from 11 subjects. Direct visualization of raw EEG signals is often unreliable due to their low signal-to-noise ratio and susceptibility to various nuisance factors, such as sensor noise, power-line interference, and inter-trial as well as inter-subject variability. These factors can distort the correlation structure and degrade the fidelity of manifold embeddings. Therefore, on the Correlation manifold, we first employ CorAtt (Hu et al., 2025), a Correlation manifold attention network, to extract more stable and discriminative representations, and then perform manifold-aware dimensionality reduction on the resulting intermediate geometric features from Subject 11. Specifically, we collect 100 feature samples evenly from 5 stimulus classes (20 samples per class), where each sample is a $15 \times 15$ Correlation matrix.

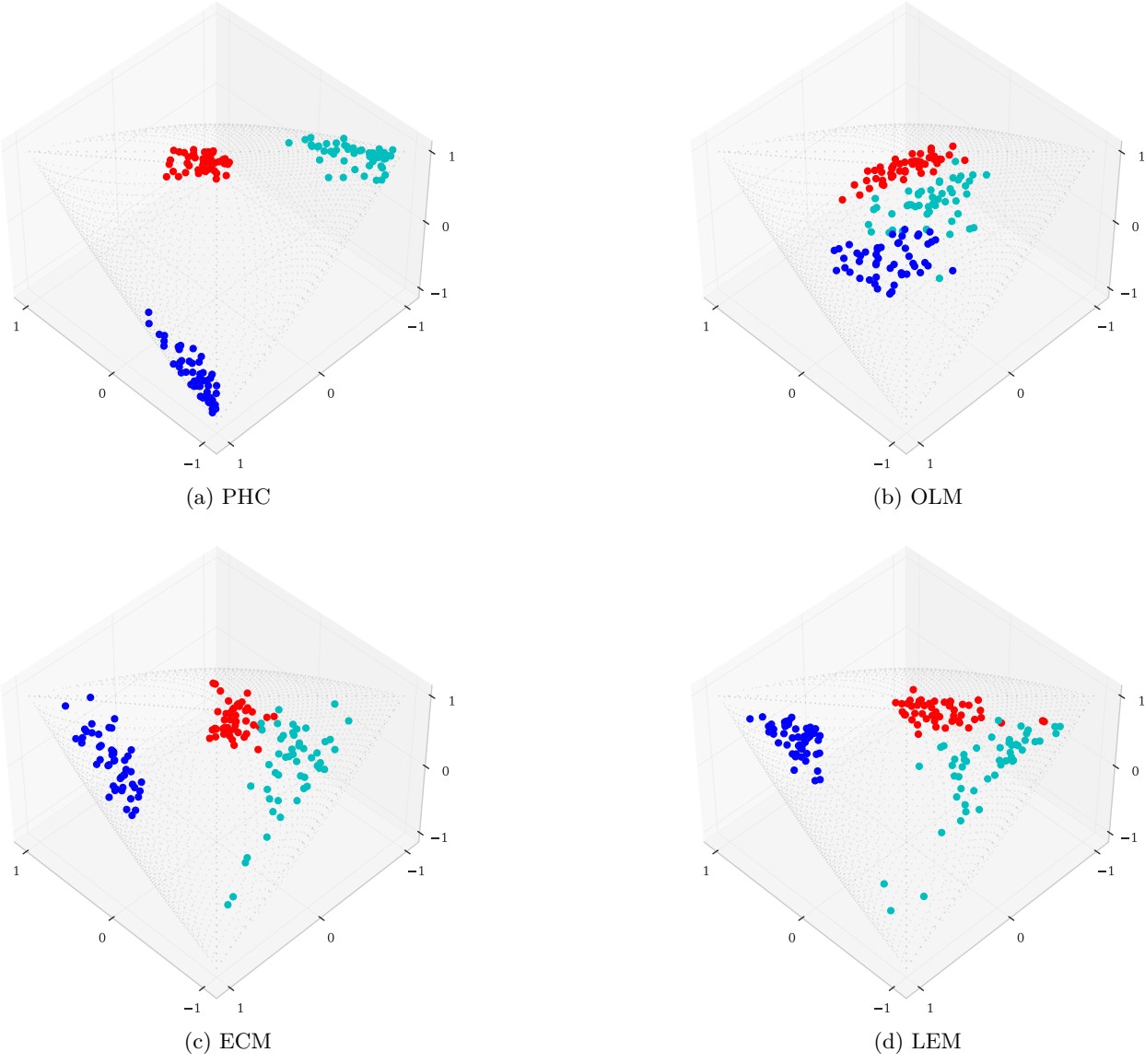

(a) PHC

(b) OLM

(c) ECM

(d) LEM

Figure 7: The impact of different Riemannian distances on the visualization of Correlation matrices on a synthetic dataset.

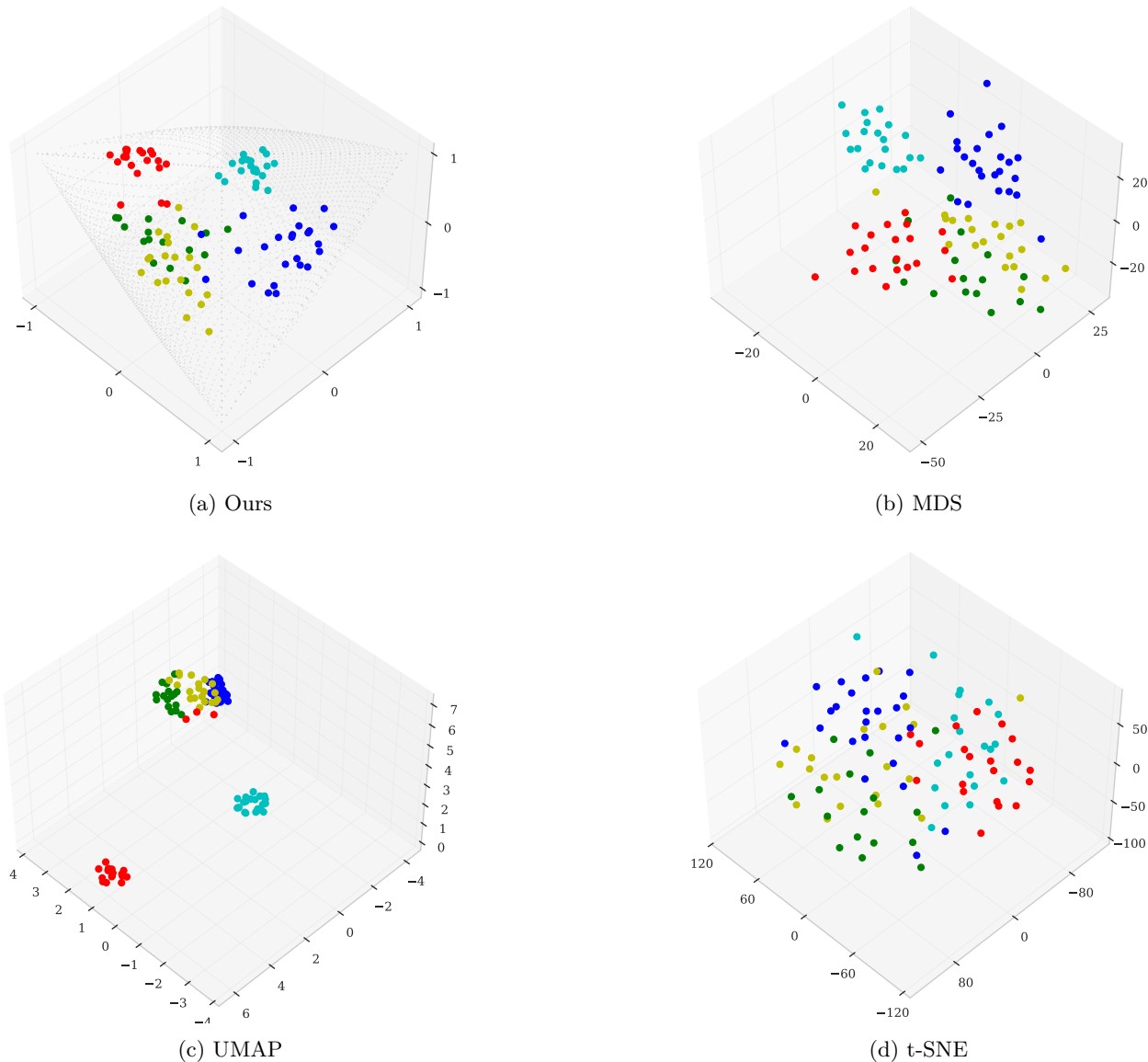

(a) Ours

(b) MDS

(c) UMAP

(d) t-SNE

Figure 8: Visualization results of the manifold-valued features from an EEG network under different methods.

The visualization results obtained by different dimensionality reduction methods are presented in Fig. 8. It can be seen that our method yields compact and well-separated clusters across the five classes, indicating that the class-discriminative structure of the manifold-valued features is effectively preserved. Although UMAP and MDS also exhibit a certain degree of class separation, they typically treat each Correlation matrix as an ordinary data point in Euclidean space, either by directly vectorizing the matrix or by constructing distances in the vectorized space. In consequence, they do not explicitly account for intrinsic matrix constraints and geometry (*e.g.*, symmetry and the manifold properties induced by Correlation matrices), which may lead to distortions of the underlying data structure in the low-dimensional embedding.

### 5.3 Experiments on the SPSD manifold

**Setup.** To the best of our knowledge, there currently exist no dedicated visualization methods specifically designed for the SPSD matrices. Since SPSD matrices and SPD matrices share closely related geometric structures, the SPD-SNE (de Surrel et al., 2025) is adopted as a reasonable surrogate baseline. In this case,

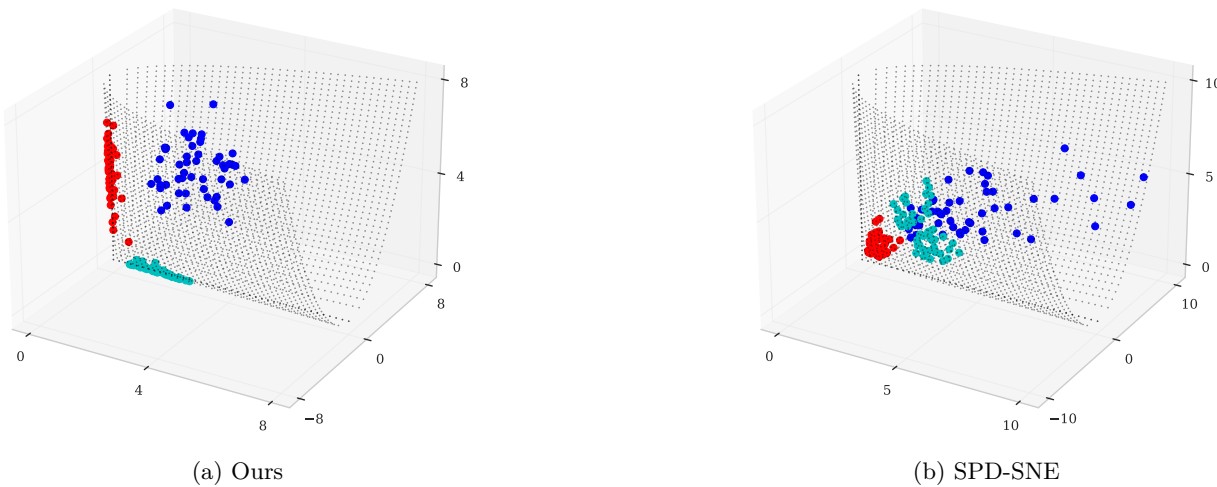

(a) Ours       (b) SPD-SNE

Figure 9: Visualizing a set of synthesized data points using the proposed method and SPD-SNE.

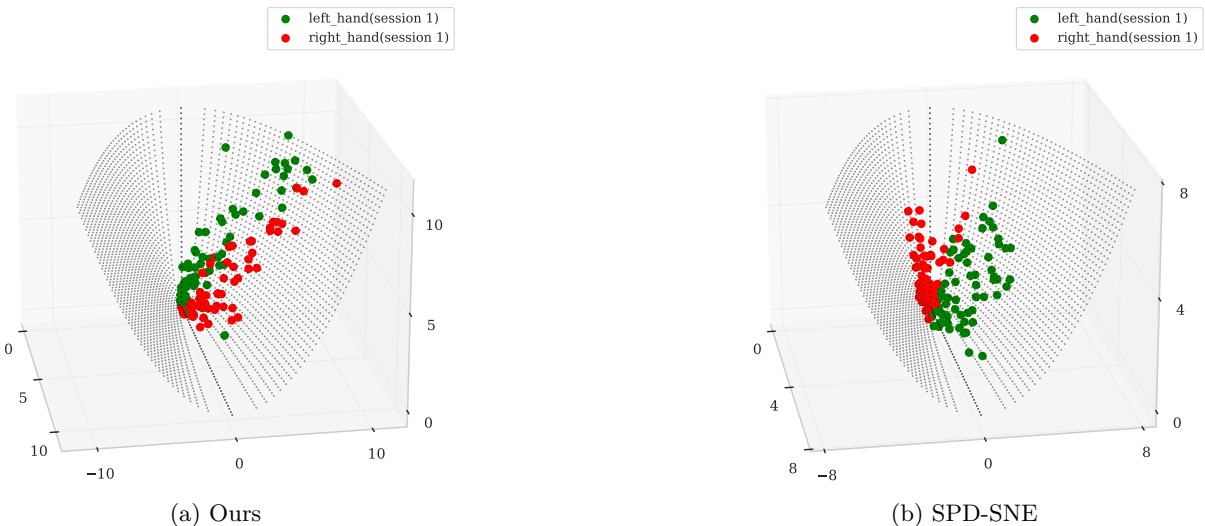

(a) Ours       (b) SPD-SNE

Figure 10: Visualizing EEG signal data using our method and SPD-SNE.

we impose a widely used regularization method on the aforementioned covariance matrix $\Sigma$, *i.e.*, $\Sigma + \epsilon I$, to produce the required SPD data points. In parallel, we generate the corresponding SPSD samples via Eigenvalue Decomposition by retaining the leading components that preserve 95% of the spectral energy. In addition, we use the official implementation provided in de Surrel et al. (2025) with default parameter settings. To ensure a fair comparison, our method is configured identically to that in the previous experiments.

**Synthetic experiments.** On the SPSD manifold, we also conduct a synthetic experiment in a 10-dimensional setting, where each data point is represented as a $4 \times 4$ SPSD matrix. Three clusters are generated, each containing 50 samples drawn from a Riemannian normal distribution with an intra-class variance of 0.4. The visualization results are shown in Fig. 9.

Both methods produce visually meaningful embeddings. However, they exhibit distinct distribution patterns due to the different geodesic structures they employ. To be specific, SPD-SNE tends to embed data points toward the interior of the positive definite cone, enforcing strict SPD constraints and resulting in clusters concentrated away from the boundary. In contrast, our proposed method distributes data points close to

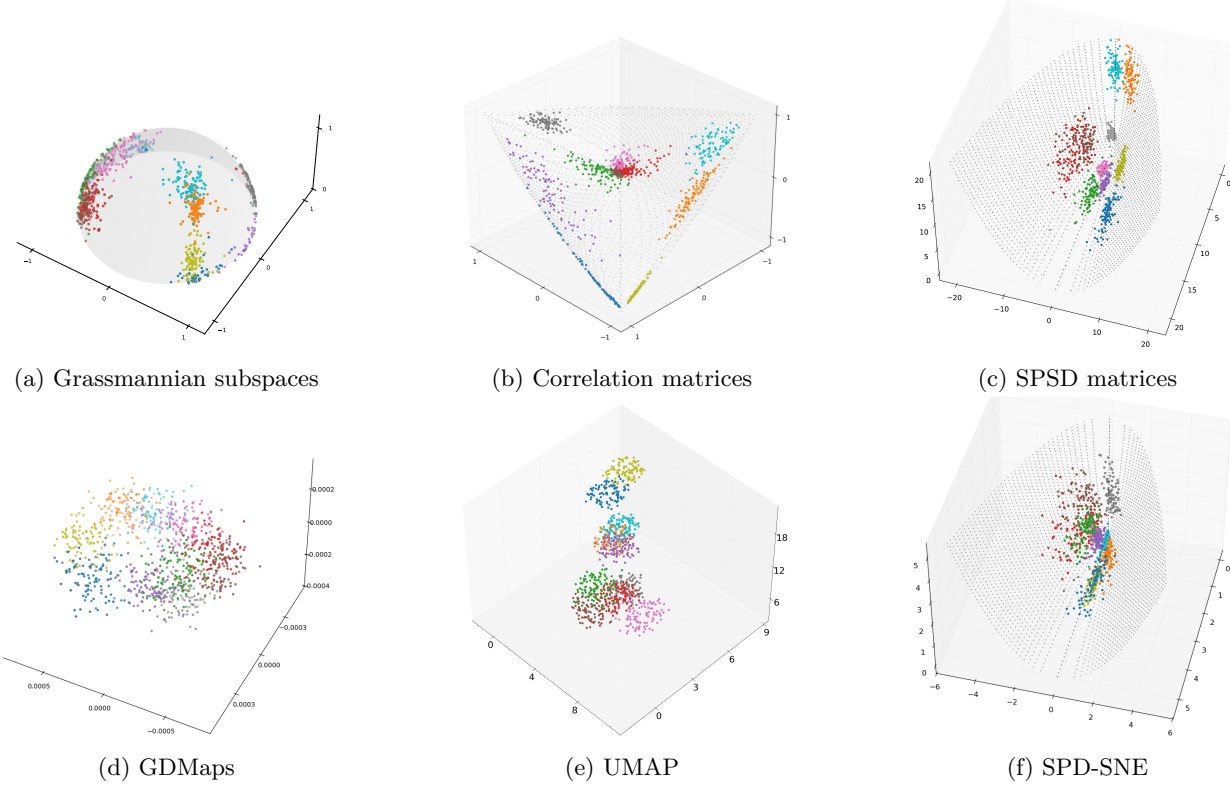

(a) Grassmannian subspaces  (b) Correlation matrices  (c) SPSD matrices

(d) GDMaps  (e) UMAP  (f) SPD-SNE

Figure 11: Comparison of different manifold visualization methods on CIFAR-10. The first row presents the low-dimensional Grassmannian subspaces, Correlation matrices, and SPSD matrices produced by our proposed method, whereas the second row displays the corresponding representations obtained by the three comparative methods.

the cone's boundary, thereby respecting the intrinsic geometry of SPSD matrices and better preserving the structural characteristics associated with rank-deficient or nearly singular samples.

**EEG signal data.** To the best of our knowledge, there are currently no established SPSD manifold neural architectures for EEG representation learning that can serve as a valid baseline. As a result, we directly apply the proposed visualization method to the raw EEG data for fair and reproducible evaluation. (Tangermann et al., 2012). Specifically, we visualized the raw EEG data of Subject 8 in the BNCI2014001 dataset (Tangermann et al., 2012), which is consistent with de Surrel et al. (2025). The experimental results are shown in Fig. 10.

It can be seen that both methods achieve a clear separation between the two categories. Note that the two visualizations are not directly comparable in a strict sense, since SPD-SNE assumes full-rank SPD inputs, whereas our method operates on low-rank SPSD matrices. This difference in matrix constraints and underlying geometry naturally leads to different embeddings. Importantly, in the low-rank regime, our method preserves local neighborhoods more faithfully, yielding a higher trustworthiness coefficient (Fig. **??**).

This improvement is expected because the low-rank SPSD construction suppresses eigen directions dominated by noise, which otherwise introduce instability into distance-based neighborhood relations. As a result, the embeddings better preserve local neighborhood structure, leading to higher trustworthiness scores.

### 5.4 Scalability Experiments on CIFAR-10 and Tiny-ImageNet

To assess the scalability of the proposed method, we make further visualization experiments on two widely used image benchmarks, *CIFAR-10* (Krizhevsky et al., 2009) and *Tiny-ImageNet* (Deng et al., 2009).

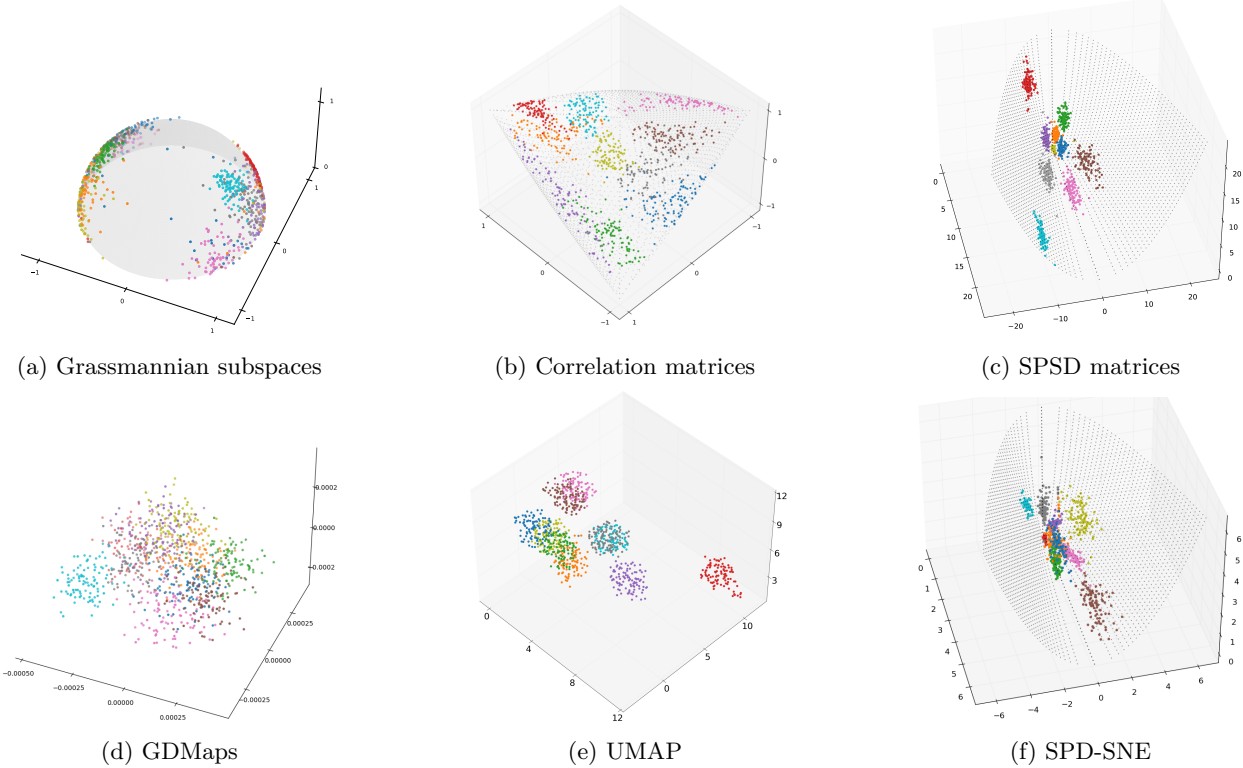

(a) Grassmannian subspaces      (b) Correlation matrices      (c) SPSD matrices

(d) GDMaps      (e) UMAP      (f) SPD-SNE

Figure 12: Comparison of different manifold visualization methods on Tiny-ImageNet. The first row presents the low-dimensional Grassmannian subspaces, Correlation matrices, and SPSD matrices produced by our proposed method, whereas the second row displays the corresponding representations obtained by the three comparative methods.

**Data preprocessing.** For each dataset, we randomly select 10 classes, with each consisting of 100 samples. This results in 1,000 samples in total. Within each class, we randomly chose 50 images and then vectorized them, followed by the utilization of PCA to obtain a new feature matrix of size $50 \times 32$. According to the construction pipeline described in Section 4.1, each matrix is further transformed into three types of matrix manifold-valued representations, given below:

- **Grassmannian subspace (with the size of $32 \times 25$)**: constructed using the top 25 right singular vectors of the covariance matrix.

- **Correlation matrix (with the size of $32 \times 32$)**: obtained by normalizing the covariance representation of the feature matrix by Eq. 2.

- **SPSD matrix (with the size of $32 \times 32$)**: approximated by truncated eigendecomposition while retaining 90% of the total spectral energy, thereby preserving the positive semi-definite structure.

**Results.** As illustrated in Fig. 11 and Fig. 12, the proposed method produces well-separated clusters across all three manifold-valued representations on both the CIFAR-10 and Tiny-ImageNet datasets. Traditional Euclidean-based methods often overemphasize nearest-neighbor attraction when processing high-dimensional matrices, which may lead to class overlap and the loss of intra-cluster density information. In contrast, our proposed method more effectively preserves both the intrinsic global topological structure and the local class distributions.

Specifically, Grassmannian subspaces and Correlation matrices effectively capture second-order statistical dependencies inherent in deep feature representations. Meanwhile, the SPSD representation exhibits strong robustness in preserving discriminative information, particularly under rank-deficient conditions.

In contrast, baseline methods such as GDMaps and SPD-SNE demonstrate limited discriminability on the more challenging Tiny-ImageNet benchmark. As intra-class variance and semantic complexity increase, the specific metric formulations adopted by these methods become less effective in capturing fine-grained semantic boundaries, resulting in noticeable overlap among visually similar categories. In particular, GDMaps relies on pre-defined kernels (e.g., Projection or Binet–Cauchy kernels) to approximate manifold affinities, which may lack sufficient sensitivity to model complex data distributions.

On the other hand, SPD-SNE inherently assumes symmetric matrices with positive definiteness, which makes it theoretically unsuitable for handling the rank-deficient structures that naturally arise in our setting. Enforcing such matrices onto the SPD manifold via a commonly used regularization strategy, i.e., by adding a small perturbation to the main diagonal part, may distort the intrinsic geometry, particularly near the boundary of the positive semi-definite cone, thereby degrading class separability. To explicitly quantify the degree of proximity to rank deficiency on the CIFAR-10 and Tiny-ImageNet datasets, we evaluate the condition number defined as $\kappa(A) = \frac{\sigma_{\max}(A)}{\sigma_{\min}(A)}$, where $\sigma_{\max}(A)$ and $\sigma_{\min}(A)$ denote the largest and smallest singular values of the SPSD matrix $A$, respectively. A larger value of $\kappa(A)$ indicates that the matrix is closer to being rank-deficient. As summarized in Table 2, a substantial proportion of the matrices in both datasets are highly ill-conditioned.

Table 2: Proportion of ill-conditioned matrices

| Dataset | $\kappa > 5 \times 10^2$ | $\kappa > 1 \times 10^3$ | $\kappa > 1.5 \times 10^3$ |
|---|---|---|---|
| CIFAR-10 | 100.00% | 91.30% | 62.10% |
| Tiny-ImageNet | 100.00% | 90.80% | 29.70% |

For dimensionality reduction under such ill-conditioned settings, our method achieves improved intra-class compactness and inter-class separability (as shown in Figs. 11c, 11f, 12c, and 12f). This demonstrates that the proposed SPSD-SNE more effectively preserves discriminative structure when the underlying data exhibit low-rank or near-rank-deficient characteristics.

Furthermore, as shown in Figs. 11e and Fig. 12e, although UMAP is capable of forming compact clusters for well-separated categories, it tends to produce significant overlap when dealing with semantically similar classes. This limitation primarily stems from its reliance on Euclidean approximations, which inevitably fail to capture the intrinsic curvature of matrix manifolds and may mischaracterize the local topological structures.

In contrast, by explicitly operating on the appropriate Riemannian manifolds and leveraging exact geodesic distances, the proposed geometry-aware framework effectively avoids distortions induced by Euclidean assumptions. As a result, it is qualified to achieve clear inter-class separation while preserving the intrinsic manifold structure, demonstrating strong generalization ability and geometric robustness on complex visual benchmarks.

## 5.5 Quantitative Evaluation of Embedding Quality

To rigorously assess the quality of the proposed method beyond qualitative visualization, we conduct three complementary quantitative evaluations on the CIFAR-10 and Tiny-ImageNet datasets across all three manifold representations: trustworthiness, $k$-nearest neighbor (KNN) classification accuracy, and Spearman's rank correlation coefficient. These metrics provide a multi-scale characterization of embedding quality, covering local neighborhood fidelity, label-neighborhood consistency, and global distance-structure preservation.

### 5.5.1 Trustworthiness

**Definition.** The trustworthiness $T(k)$ at neighborhood size $k$ quantifies the proportion of $k$-nearest neighbors in the low-dimensional embedding that are also genuine neighbors in the original high-dimensional

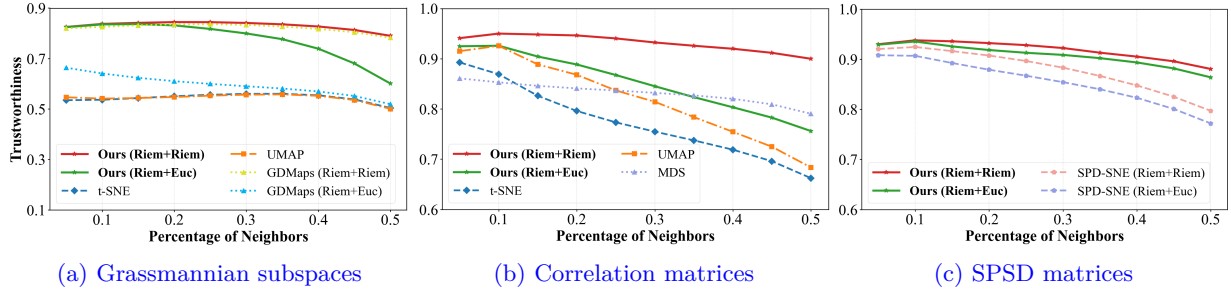

(a) Grassmannian subspaces      (b) Correlation matrices      (c) SPSD matrices

Figure 13: Trustworthiness evaluation on CIFAR-10 under three manifold-valued data representations: (a) Grassmannian subspaces, (b) correlation matrices, and (c) SPSD matrices.

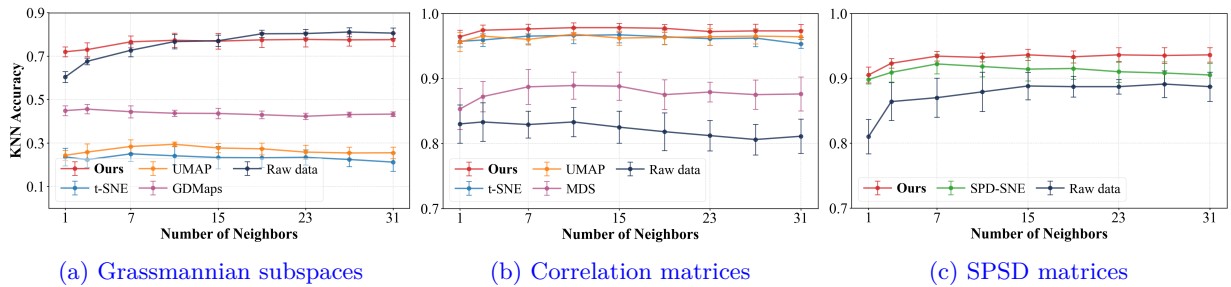

(a) Grassmannian subspaces      (b) Correlation matrices      (c) SPSD matrices

Figure 14: KNN classification accuracy on CIFAR-10 under three manifold-valued data representations: (a) Grassmannian subspaces, (b) correlation matrices, and (c) SPSD matrices.

space, thereby penalizing false neighbors introduced by the embedding. Following Venna & Kaski (2006):

$$T(k) = 1 - \frac{2}{Nk(2N - 3k - 1)} \sum_{i=1}^{N} \sum_{j \in \mathcal{U}_k(i)} \big( r(i, j) - k \big) \tag{16}$$

where $N$ is the number of data points, $r(i, j)$ denotes the rank of point $j$ among all points ordered by distance to point $i$ in the original high-dimensional space, and $\mathcal{U}_k(i)$ is the set of points that are among the $k$-nearest neighbors of $i$ in the low-dimensional embedding but not in the original space. The score lies in $[0, 1]$, with higher values indicating fewer false neighbors and better local structure preservation.

**Distance Measurement Protocol.** A critical challenge in evaluating manifold-valued embeddings is the choice of the distance metric used to define local "neighborhoods". Evaluating a Riemannian embedding method solely with Euclidean distances may introduce metric mismatch, while evaluating it only with intrinsic Riemannian distances may raise concerns about metric circularity. To ensure a fair and rigorous evaluation, we establish the following measurement protocol:

- **High-Dimensional Ground Truth:** For all evaluations, high-dimensional pairwise distances are computed using the intrinsic Riemannian metric appropriate for the data (i.e., the specific metrics defined for the Grassmann, Correlation, and SPSD manifolds). This ensures that the reference neighborhood structure reflects the underlying manifold geometry.

- **Low-Dimensional Evaluation (Euclidean baselines):** For traditional methods that map data to a flat space (e.g., t-SNE, UMAP, MDS), nearest neighbors in the embedding space are computed using standard Euclidean distance.

- **Low-Dimensional Evaluation (Riemannian methods):** For our proposed framework and other manifold-aware baselines (e.g., GDMaps, SPD-SNE), we report two complementary evaluation protocols:

– *Intrinsic Evaluation (Riem+Riem):* Neighbors are determined using the corresponding Riemannian distance on the low-dimensional target manifold. This protocol evaluates how well the embedding preserves intrinsic manifold neighborhoods under the geometry assumed by the representation.

– *Naïve Evaluation (Riem+Euc):* Neighbors are computed using Euclidean distance directly on the low-dimensional embedding coordinates. This serves as a robustness check to determine whether the learned representations preserve meaningful neighborhood structure even when downstream applications (e.g., standard classifiers or clustering algorithms) naïvely ignore the intrinsic manifold geometry.

**Results.** The trustworthiness curves are shown in Fig. 13, with detailed quantitative results provided in Tab. 7 in Appendix C.1. Across all evaluated matrix manifolds (Grassmann, Correlation, and SPSD), manifold-aware methods (such as our proposed framework, GDMaps) generally outperform traditional Euclidean baselines. The fundamental reason for this performance gap lies in geometric mismatch: Euclidean methods rely on flat-space distances and therefore may not fully capture the intrinsic curvature and matrix constraints of the data. Consequently, traditional baselines exhibit different degradation patterns. UMAP, for instance, exhibits competitive trustworthiness at highly localized scales (small $k$) but deteriorates rapidly as the neighborhood expands on the Correlation manifold (see Fig. 13b). This reflects UMAP's strong algorithmic bias toward preserving immediate local point cloud connectivity while being less effective at preserving broader intrinsic relationships on curved spaces. Conversely, MDS is designed to prioritize global distance preservation; it accordingly yields relatively stable but persistently lower trustworthiness scores across all scales, lacking the sensitivity to capture fine-grained local manifold intricacies. t-SNE also obtains lower trustworthiness in this setting, suggesting that Euclidean-based affinity construction may be insufficient for preserving intrinsic Riemannian neighborhoods.

Furthermore, our dual-evaluation protocol highlights the critical importance of proper metric selection when quantifying manifold embeddings. Our method, when evaluated under the intrinsic Riem+Riem protocol, achieves the highest trustworthiness overall and remains stable even as the neighborhood size approaches $k/N = 0.50$. However, a pronounced divergence emerges under the extrinsic Riem+Euc protocol: trustworthiness remains highly competitive at very small neighborhood scales but declines sharply as the neighborhood radius increases. This phenomenon is theoretically consistent with the mathematical definition of smooth manifolds. Within a highly localized region, a Riemannian manifold can be locally approximated by a flat Euclidean space (i.e., its tangent space), allowing Euclidean distances to serve as reasonable proxies for true geodesics. However, as the neighborhood size expands beyond the validity of this local linear approximation, the intrinsic curvature invalidates Euclidean metrics.

### 5.5.2 KNN Classification Accuracy

**Setup.** To evaluate the practical utility of the dimensionality reduction, we use KNN classification accuracy as a label-based downstream metric. The significance of this approach is that it evaluates whether the embedding preserves class-relevant neighborhood structure. Because it relies on class labels, it provides an independent downstream evaluation of class-label separability and avoids relying solely on the distance metric used by the embedding objective.

Given a dataset of $N$ samples with ground-truth labels $y_i$, the KNN accuracy evaluates the proportion of samples correctly classified by a majority vote among their $k$ nearest neighbors in the embedding space (Cover & Hart, 1967). Mathematically, it is defined as:

$$\text{Accuracy}(k) = \frac{1}{N} \sum_{i=1}^{N} \mathbb{I}(y_i = \hat{y}_i^{(k)}) \tag{17}$$

where $\mathbb{I}(\cdot)$ is the indicator function, and $\hat{y}_i^{(k)}$ is the predicted label determined by a majority vote among the $k$ nearest neighbors of sample $i$ in the low-dimensional embedding space. For reference, we also report the KNN accuracy directly computed on the original high-dimensional manifold-valued data using the corresponding intrinsic Riemannian distances. For KNN evaluation, we adopt a low-dimensional distance

protocol similar to that used for the trustworthiness evaluation: Euclidean distance is used for flat embeddings, while Riemannian embeddings are evaluated using intrinsic Riemannian distance. The evaluation results are presented in Fig. 14 and Tab. 8.

**Results.** Across all evaluated matrix manifolds (Grassmann, Correlation, and SPSD), our proposed Riemannian framework consistently yields the highest KNN classification accuracy compared to both Euclidean baselines and existing manifold-aware methods. Traditional dimensionality reduction techniques (t-SNE, UMAP, and MDS) generally exhibit substantial performance degradation, most notably on the Grassmann manifold, where their accuracies fall below 0.300. This suggests that relying on Euclidean approximations to compute affinities structurally distorts the discriminative boundaries inherent to the original data.

On both the Correlation and SPSD manifolds, the low-dimensional embeddings generated by our approach generally yield higher accuracy than the raw high-dimensional baselines across varying neighborhood sizes. This phenomenon indicates that raw high-dimensional covariance or semi-definite structures may contain redundant information or geometric noise. By structurally projecting the data onto an appropriate lower-dimensional Riemannian manifold, our method effectively filters out this task-irrelevant variance, which may contribute to improved KNN classification accuracy.

Furthermore, our approach maintains a steady advantage over other manifold-aware baselines, such as SPD-SNE on the SPSD manifold, and this performance difference may be related to the underlying metric choice: our quotient-geometric formulation is naturally suited to handle the rank-deficient SPSD matrices.. Overall, these findings support the practical utility of geometry-aware dimensionality reduction in exposing the semantic separability of matrix-valued data.

### 5.5.3 Spearman's Rank Correlation Coefficient

Table 3: Quantitative evaluation of global structure preservation using Spearman's rank correlation coefficient ($\rho_s$) on the CIFAR-10 and Tiny-ImageNet dataset.

(a) CIFAR-10

| Manifold | Method | $\rho_s$ |
|---|---|---|
| Grassmann manifold | Ours | **0.6081** |
| | GDMaps | 0.6047 |
| | t-SNE | 0.0148 |
| | UMAP | 0.0044 |
| Correlation manifold | Ours | **0.8442** |
| | MDS | 0.6746 |
| | t-SNE | 0.4644 |
| | UMAP | 0.4363 |
| SPSD manifold | Ours | **0.7904** |
| | SPD-SNE | 0.6553 |

(b) Tiny-ImageNet

| Manifold | Method | $\rho_s$ |
|---|---|---|
| Grassmann manifold | Ours | **0.4383** |
| | GDMaps | 0.4187 |
| | t-SNE | 0.0023 |
| | UMAP | 0.0025 |
| Correlation manifold | Ours | **0.8182** |
| | MDS | 0.5551 |
| | t-SNE | 0.3877 |
| | UMAP | 0.5085 |
| SPSD manifold | Ours | **0.7758** |
| | SPD-SNE | 0.7119 |

**Setup.** While trustworthiness and KNN accuracy evaluate the fidelity of local neighborhoods and semantic boundaries, they do not directly measure whether the global ordering of pairwise distances is preserved. To assess global geometric structure, we use Spearman's rank correlation coefficient ($\rho_s$) (Spearman, 1961). Trustworthiness and KNN focus primarily on local neighborhood preservation and local semantic separability, while Spearman's rank correlation evaluates whether the embedding preserves the global ordering of pairwise intrinsic distances. For all $M = \binom{N}{2}$ unique point pairs, let $R_{ij}^{\text{high}}$ and $R_{ij}^{\text{low}}$ denote the ranks of the pairwise distances between points $i$ and $j$ in the high-dimensional and low-dimensional spaces, respectively. The

coefficient is formulated as:

$$\rho_s = 1 - \frac{6 \sum\limits_{(i,j)} \left( R_{ij}^{\text{high}} - R_{ij}^{\text{low}} \right)^2}{M(M^2 - 1)}. \tag{18}$$

A value of $\rho_s$ close to 1 indicates strong monotonic preservation of global pairwise distance orderings. Adopting the distance measurement protocol outlined in Sec 5.5.1, this metric provides an objective quantification of how well the global geometric structure is retained after dimensionality reduction. Results for both the CIFAR-10 and Tiny-ImageNet datasets are summarized in Tab. 3.

**Results.** The empirical results indicate that the proposed Riemannian framework generally yields higher global rank correlations across the evaluated matrix manifolds (Grassmann, Correlation, and SPSD) on both datasets. Traditional Euclidean-based methods, particularly t-SNE and UMAP, appear to struggle with global structure preservation in these settings. This difficulty is most evident on the Grassmann manifold, where t-SNE and UMAP yield near-zero correlations, suggesting a substantial loss of the original macroscopic geometry. Even MDS, which is explicitly designed to preserve global structures, shows a lower coefficient compared to our method (e.g., 0.6746 versus 0.8442 on the Correlation manifold for CIFAR-10). These observations suggest that methods optimized under flat Euclidean assumptions may not fully capture the global structure of non-Euclidean matrix manifolds.

Furthermore, our approach exhibits favorable performance when compared to existing manifold-aware baselines. For instance, on the SPSD manifold, our method yields higher coefficients than SPD-SNE across both datasets. Overall, the results suggest that incorporating intrinsic manifold metrics helps mitigate macroscopic distortions, facilitating a better balance between local neighborhood fidelity and global geometric coherence.

## 6 Conclusion

In this paper, we introduce a visualization framework for high-dimensional matrix-valued data defined on Riemannian manifolds, with particular emphasis on the Grassmannian subspaces, Correlation matrices, and SPSD matrices. By explicitly incorporating intrinsic geodesic distances and geometry-aware probability distributions, the proposed method effectively preserves both local and global geometric structures during dimensionality reduction. Extensive experimental results on synthetic data points, benchmarking datasets, and manifold network features verify that our method consistently outperforms Euclidean geometry-based baselines as well as existing manifold-specific methods in capturing geometric relationships of manifold data.

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

# A   Appendix

```
from grass_imp import Grass_SNE
from cor_imp import Cor_SNE
from spsd_imp import SPSD_SNE
# Visualization: Correlation matrices, Grassmann suspaces, and SPSD matrices
embedding_h, labels, size = get_data()
# Correlation matrices implemented metrics: ECM, LEM, PHC, OLM
# Grassmannian implemented metrics: Geodesic, Asimov, Binet Cauchy, Chordal, Fubini
    Study, Martin, Procrustes, Projection, Spectral
# SPSD matrices implemented metrics: Wasserstein
sne_embedding = run_SNE(embedding_h, manifold='Grassmannian', metric='Asimov', shape=
    size, t_gamma=1, learning_rate=0.05, max_iter=10000)
plot_low_dim(sne_embedding, labels)
```

Figure 15: A concise demo of using our proposed method

## A.1   Distance metrics on Grassmannian manifolds

| Metric | Riemannian distance |
|--------|---------------------|
| Chordal | $\left(\sum_{i=1}^{k} \sin^2 \theta_i\right)^{1/2}$ |
| Martin | $\left(\log \prod_{i=1}^{k} 1/\cos^2 \theta_i\right)^{1/2}$ |
| Projection | $\sin \theta_{max}$ |

Table 4: Distances on the Grassmann manifold (Ye & Lim, 2016).

**Principal angles** are a set of angles that measure the angle between two subspaces. They are calculated starting from the "closest" direction and can fully describe the relative position of the two subspaces. It can be defined as follows. Let $U_1, U_2 \subset \mathbb{R}^d$ be two subspaces with dimensions $\dim(U_1) = k$ and $\dim(U_2) = l$, respectively. Define $r = \min(k, l)$. The principal angles between $U_1$ and $U_2$ are denoted by

$$0 \le \theta_1 \le \theta_2 \le \cdots \le \theta_r \le \frac{\pi}{2},$$

and are defined recursively as follows.

For each $i = 1, \ldots, r$, the $i$-th principal angle $\theta_i$ is defined via the optimization problem:

$$\cos \theta_i = \max_{\substack{p \in U_1, \|p\|=1, p \perp p_1, \ldots, p_{i-1} \\ q \in U_2, \|q\|=1, q \perp q_1, \ldots, q_{i-1}}} p^\top q, \tag{19}$$

where $(p_j, q_j)$ are the principal vectors associated with the $j$-th principal angle $\theta_j$ for $j < i$.

Geometrically, each $\theta_i$ represents the smallest possible angle between directions in $U_1$ and $U_2$ that are orthogonal to the previously identified principal vectors. Collectively, the principal angles fully characterize the relative positions of the two subspaces up to rotation, and they can be computed via SVD.

Let $U_1 \in \mathbb{R}^{d \times k}$ and $U_2 \in \mathbb{R}^{d \times l}$ be matrices whose columns form orthonormal bases for $U_1$ and $U_2$, respectively. The principal angles $\theta_1, \ldots, \theta_r$ can be efficiently computed from SVD of the matrix $U_1^\top U_2 \in \mathbb{R}^{k \times l}$. Specifically, let

$$U_1^\top U_2 = W \Sigma Z^\top,$$

where $\Sigma = \mathrm{diag}(\sigma_1, \ldots, \sigma_r)$, with singular values $\sigma_1 \ge \cdots \ge \sigma_r \in [0, 1]$. Then the principal angles are given by:

$$\theta_i = \cos^{-1}(\sigma_i), \quad i = 1, \ldots, r. \tag{20}$$

The associated principal vectors are given by the columns of $U_1 W$ and $U_2 Z$, corresponding to the left and right singular vectors of $U_1^\top U_2$, respectively.

Table 4 shows different distance metrics based on principal angles on the Grassmann manifold.

On Grassmann manifolds, common subspace distances include Chordal distance, Martin distance, and Projection distance. All three can characterize the relative positions between subspaces using principal angles, but they differ in the geometric features they focus on. Chordal distance considers all principal angles, equivalent to comparing the overall difference between the projection matrices of two subspaces; it is numerically stable and insensitive to noise. Projection distance is determined only by the largest principal angle, characterizing the degree of deviation between two subspaces in the direction of greatest inconsistency, suitable for worst-case scenarios or scenes requiring strong discriminative power. In contrast, Martin distance, through a nonlinear transformation of the principal angles, emphasizes the differences when subspaces are nearly orthogonal, is more sensitive to geometric changes, but is relatively complex in numerical computation.

For more details, please refer to Ye & Lim (2016).

### A.2 Distance metrics on Correlation manifolds

| Metric | Riemannian Distance |
|--------|---------------------|
| ECM | $\|\phi(C_2) - \phi(C_1)\|_F$ |
| LEM | $\|\log(\phi(C_2)) - \log(\phi(C_1))\|_F$ |
| OLM | $\|\mathrm{Off}\left(\log(C_2) - \log(C_1)\right)\|_F$ |

Table 5: Distances on the Correlation manifold (Thanwerdas & Pennec, 2022; Thanwerdas, 2024).

The Euclidean-Cholesky metric (ECM) and the log-Euclidean-Cholesky metric (LEM) are both constructed by mapping Correlation matrices to a lower triangular matrix space via the Cholesky decomposition, equipping this space with a Euclidean structure, and pulling the metric back to the original manifold. Both metrics are flat and geodesically complete, ensuring the uniqueness of the Fréchet mean. ECM directly uses a Euclidean metric on the Cholesky factors, leading to simple and efficient computations, whereas LEM applies a logarithmic transform in the Cholesky domain, yielding a scale-adaptive geometry at a slightly higher computational cost. In contrast, off-log metrics (OLM) rely on a log-Euclidean-type parametrization that separates diagonal and off-diagonal components without using a triangular factorization. OLMs preserve permutation invariance and also induce a flat geometry, but differ from ECM and LEM in both interpretation and numerical behavior. The definitions of these three distances are as follows:

**The ECM distance.** Let $C \in \mathrm{Cor}^+(n)$ be a full-rank Correlation matrix, and denote its normalized Cholesky factor by

$$\phi(C) = \mathrm{Diag}(\mathrm{Chol}(C))^{-1}\mathrm{Chol}(C) \in \mathrm{LT}_1(n), \tag{21}$$

where $\mathrm{Chol}(C) \in \mathrm{LT}_+(n)$ is the unique lower triangular matrix with positive diagonal such that $C = \mathrm{Chol}(C) \cdot \mathrm{Chol}(C)^\top$, and $\mathrm{LT}_1(n)$ denotes the set of unit-diagonal lower triangular matrices.

The ECM distance between two Correlation matrices $C_1, C_2 \in \mathrm{Cor}^+(n)$ is defined as

$$d_{\mathrm{EC}}(C_1, C_2) = \|\phi(C_1) - \phi(C_2)\|_F, \tag{22}$$

where $\|\cdot\|_F$ denotes the Frobenius norm. This distance corresponds to the Euclidean norm in the space $\mathrm{LT}_1(n)$.

**The LEM distance.** Let $\log : \mathrm{LT}_1(n) \to \mathrm{LT}_0(n)$ denote the matrix logarithm, which is a diffeomorphism from the Lie group of unit-diagonal lower triangular matrices $\mathrm{LT}_1(n)$ to its Lie algebra $\mathrm{LT}_0(n)$, the space of strictly lower triangular matrices. The LEM distance between two Correlation matrices $C_1, C_2 \in \mathrm{Cor}^+(n)$ is defined by

$$d_{\mathrm{LE}}(C_1, C_2) = \|\log\left(\phi(C_1)\right) - \log\left(\phi(C_2)\right)\|_F, \tag{23}$$

where the logarithm is computed in the matrix sense. This metric corresponds to the Euclidean distance in the logarithmic coordinates of the Lie algebra $\mathrm{LT}_0(n)$.

Next, we define the off-log distance as the geodesic distance induced by the off-log metric (OLM) on the manifold of full-rank Correlation matrices $\mathrm{Cor}^+(n)$.

**The OLM distance.** Let $\mathrm{Log} : \mathrm{Cor}^+(n) \to \mathrm{Hol}(n)$ be the off-log diffeomorphism, given by

$$\mathrm{Log}(C) := \mathrm{Off}(\log(C)), \tag{24}$$

where $\mathrm{Hol}(n)$ denotes the space of symmetric matrices with vanishing diagonal:

$$\mathrm{Hol}(n) := \{X \in \mathrm{Sym}(n) \mid \mathrm{Diag}(X) = 0\}. \tag{25}$$

Given two Correlation matrices $C_1, C_2 \in \mathrm{Cor}^+(n)$, the off-log distance between them is defined by

$$d_{\mathrm{OL}}(C_1, C_2) := \|\mathrm{Log}(C_2) - \mathrm{Log}(C_1)\|_{\mathrm{Hol}}, \tag{26}$$

where $\|\cdot\|_{\mathrm{Hol}}$ is the norm induced by a permutation-invariant inner product on $\mathrm{Hol}(n)$. A commonly used instance is the Frobenius norm:

$$\|X\|_{\mathrm{Hol}}^2 = \mathrm{tr}(X^2), \quad \forall X \in \mathrm{Hol}(n),$$

which leads to the following closed-form expression:

$$d_{\mathrm{OL}}(C_1, C_2) = \|\mathrm{Off}\left(\log(C_2) - \log(C_1)\right)\|_F. \tag{27}$$

This distance is smooth, flat, and invariant under permutations of variables.

These three distance metrics are shown in Table 5. For more details, please refer to Thanwerdas & Pennec (2022); Thanwerdas (2024)

### A.3 Derivation of gradients

We now compute the gradient of the loss function with respect to each embedded point on the Grassmann manifold. Recall that the objective function is the Kullback–Leibler divergence between the pairwise similarities in the high-dimensional and low-dimensional spaces:

$$C = \sum_i \mathrm{KL}(P \parallel Q) = \sum_i \sum_{j \neq i} p_{ij} \log \frac{p_{ij}}{q_{ij}},$$

To compute the gradient with respect to the embedding $Y_i$, we first apply the chain rule:

$$\nabla_{Y_i} C = 2 \sum_{j \neq i} \frac{\partial C}{\partial \delta_{ij}} \nabla_{Y_i} \delta_{ij},$$

**Computing $\frac{\partial C}{\partial \delta_{ij}}$**

Let the normalization factor Z be defined as:

$$Z = \sum_{m \neq n} \left(1 + \delta_{mn}^2\right)^{-1},$$

so that the low-dimensional similarity $q_{ij}$ becomes:

$$q_{ij} = \frac{(1 + \delta_{ij}^2)^{-1}}{Z}.$$

Rewriting the cost function $C$, we have:

$$C = \sum_i \sum_{j \neq i} p_{ij} \log p_{ij} - \sum_i \sum_{j \neq i} p_{ij} \log \left(\frac{(1 + \delta_{ij}^2)^{-1}}{Z}\right).$$

The first term is constant with respect to $Y_i$ and thus can be ignored in the gradient. Taking derivatives yields:

$$\frac{\partial C}{\partial \delta_{ij}} = \frac{\partial}{\partial \delta_{ij}} \left[p_{ij} \log(1 + \delta_{ij}^2)\right] + \frac{\partial}{\partial \delta_{ij}} \left[\sum_{m \neq n} p_{mn} \log Z\right].$$

The first term evaluates to:

$$\frac{\partial}{\partial \delta_{ij}} \left[p_{ij} \log(1 + \delta_{ij}^2)\right] = p_{ij} \cdot \frac{2\delta_{ij}}{1 + \delta_{ij}^2}.$$

For the second term, since $\sum_{m \neq n} p_{mn} = 1$, and noting:

$$\frac{\partial Z}{\partial \delta_{ij}} = \frac{\partial}{\partial \delta_{ij}} \left[\sum_{m \neq n} (1 + \delta_{mn}^2)^{-1}\right] = -\frac{2\delta_{ij}}{(1 + \delta_{ij}^2)^2},$$

we obtain:

$$\frac{\partial}{\partial \delta_{ij}} \left[\sum_{m \neq n} p_{mn} \log Z\right] = \frac{1}{Z} \cdot \left(-\frac{2\delta_{ij}}{(1 + \delta_{ij}^2)^2}\right) = -\frac{2\delta_{ij} q_{ij}}{1 + \delta_{ij}^2}.$$

Therefore, the full derivative becomes:

$$\frac{\partial C}{\partial \delta_{ij}} = 2\delta_{ij} \cdot \frac{p_{ij} - q_{ij}}{1 + \delta_{ij}^2}.$$

**Computing $\nabla_{Y_i} \delta_{ij}$**

Recall that the geodesic distance on the Grassmann manifold is:

$$\delta_{ij} = d(Y_i, Y_j) = \left(\sum_{l=1}^q \theta_l^2\right)^{1/2},$$

Then:

$$\nabla_{Y_i} \delta_{ij}^2 = 2 \sum_{l=1}^{q} \theta_l \nabla_{Y_i} \theta_l.$$

Since $\theta_l = \arccos(\sigma_l)$, where $\sigma_l$ is the $l$-th singular value of $Y_i^\top Y_j$, we obtain:

$$\nabla_{Y_i} \theta_l = -\frac{1}{\sqrt{1 - \sigma_l^2}} \cdot \nabla_{Y_i} \sigma_l,$$

and from matrix calculus:

$$\nabla_{Y_i} \sigma_l = u_l v_l^\top Y_j^\top,$$

where $u_l$ and $v_l$ are the $l$-th left and right singular vectors of $Y_i^\top Y_j$, respectively.

Combining these results and projecting onto the tangent space of the Grassmann manifold, we obtain:

$$\nabla_{Y_i} \delta_{ij} = (I - Y_i Y_i^\top) \left( \sum_{l=1}^{q} \frac{-\theta_l}{\sqrt{1 - \sigma_l^2}} u_l v_l^\top Y_j^\top \right).$$

**Final Expression**

Substituting the expressions of $\frac{\partial C}{\partial \delta_{ij}}$ and $\nabla_{Y_i} \delta_{ij}$, we obtain the final gradient of the cost function:

$$\nabla_{Y_i} C = 4 \sum_{j \neq i} \delta_{ij} \cdot \frac{p_{ij} - q_{ij}}{1 + \delta_{ij}^2} \cdot \nabla_{Y_i} \delta_{ij}.$$

# B    Discussion on Limitations

In this section, we will highlight several practical limitations of the proposed method that warrant further exploration.

## B.1    Memory and computational cost.

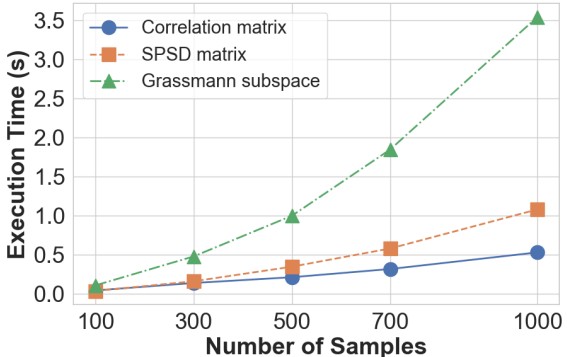

Figure 16: Runtime comparison (s/iteration) of the proposed method under three manifold-valued data types.

Table 6: Runtime comparison (s/iteration) under different matrix manifold-valued representations, sample sizes, and matrix size.

| Manifolds | Matrix size | Sample sizes | | | | |
|---|---|---|---|---|---|---|
| | | 100 | 300 | 500 | 700 | 1000 |
| Correlation matrix | $4 \times 4$ | 0.040 | 0.129 | 0.207 | 0.294 | 0.565 |
| | $8 \times 8$ | 0.040 | 0.124 | 0.215 | 0.294 | 0.549 |
| | $16 \times 16$ | 0.041 | 0.138 | 0.198 | 0.328 | 0.533 |
| | $32 \times 32$ | 0.041 | 0.136 | 0.210 | 0.315 | 0.528 |
| SPSD matrix | $4 \times 4$ | 0.035 | 0.158 | 0.336 | 0.581 | 1.068 |
| | $8 \times 8$ | 0.034 | 0.156 | 0.365 | 0.583 | 1.082 |
| | $16 \times 16$ | 0.035 | 0.155 | 0.334 | 0.581 | 1.093 |
| | $32 \times 32$ | 0.034 | 0.158 | 0.345 | 0.578 | 1.079 |
| Grassmannian subspace | $4 \times 2$ | 0.097 | 0.452 | 1.074 | 1.857 | 3.507 |
| | $8 \times 4$ | 0.103 | 0.476 | 1.128 | 1.930 | 3.693 |
| | $16 \times 8$ | 0.104 | 0.483 | 1.213 | 1.881 | 3.531 |
| | $32 \times 16$ | 0.106 | 0.477 | 0.998 | 1.846 | 3.545 |

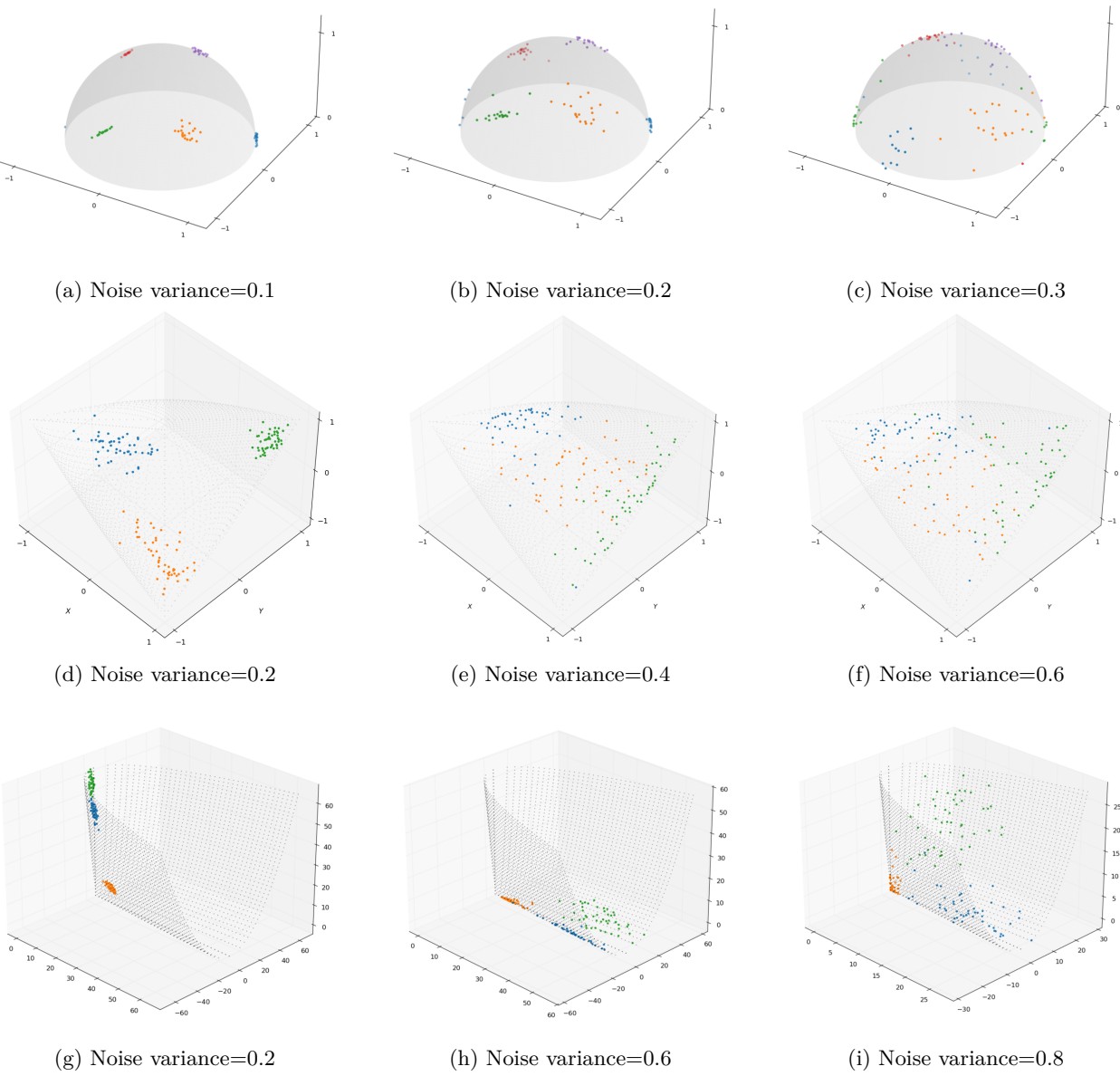

(a) Noise variance=0.1      (b) Noise variance=0.2      (c) Noise variance=0.3

(d) Noise variance=0.2      (e) Noise variance=0.4      (f) Noise variance=0.6

(g) Noise variance=0.2      (h) Noise variance=0.6      (i) Noise variance=0.8

Figure 17: The impact of different noise intensities on three manifold visualization methods. It is evident that the visualization effect deteriorates as the noise level of the data increases.

Compared to vector-based representations, matrix-manifold representations inevitably introduce substantially higher memory overhead. Even a $3 \times 3$ SPSD matrix has 6 degrees of freedom, and the storage cost increases rapidly as the matrix size grows. With the increased matrix dimension and the number of samples, memory usage will become a significant bottleneck. Furthermore, computing pairwise Riemannian distances requires $N^2$ comparisons across $N$ samples. Each evaluation may involve computationally intensive manifold-specific operations, such as eigenvalue decomposition, SVD, or Cholesky decomposition, thereby increasing the overall computational burden. More specfically, the distance computation on the Grassmann manifold relies on SVD, leading to a complexity of $O(N^2 q^2 (d + q))$, where $N$, $d$, and $q$ signify the number of samples, the ambient space dimension, and the subspace dimension, respectively. For the related Correlation and

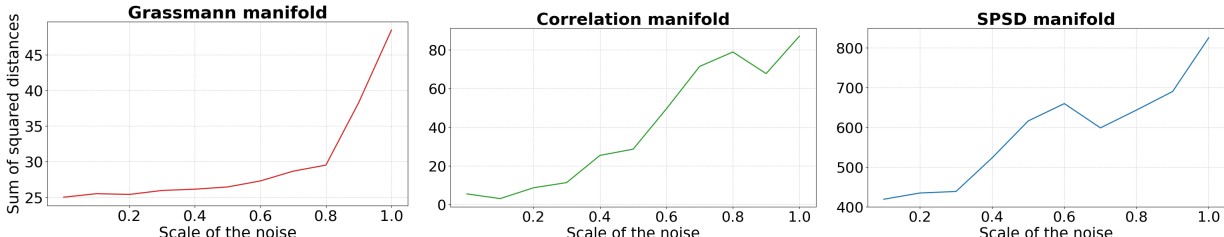

Figure 18: Dimensionality reduction results of three manifolds on synthetic datasets: Compare the distance between the reduced geodesics and the real geodesics under different noise intensities.

SPSD manifolds, the distance computations involve Cholesky decomposition and eigenvalue decomposition, resulting in the complexities of $O(Nk^3 + N^2k^2)$ and $O(N^2k^3)$, respectively. Wherein, $k$ denotes the number of rows and columns of the Correlation and SPSD matrices.

We further empirically evaluate the computational cost across all three involved matrix manifolds, with the results displayed in Fig. 16 and Tab. 6. From Fig. 16, it can be found that increasing the number of samples leads to a rapid growth in the computation time, being consistent with the theoretical analysis mentioned above. Among the three settings, the Grassmann manifold is the most computationally demanding. This is mainly because the optimization on it explicitly depends on the ambient dimension $d$, relies on repeated SVD operations that carry a heavier algorithmic burden than Cholesky-based computations, and additionally requires tangent-space projections during Riemannian optimization. By contrast, the computations and optimizations on the Correlation and SPSD manifolds are relatively more efficient. Additionally, the detailed results in Tab. 6 reveal that while runtime is predominantly driven by the sample size, the impact of matrix dimensions remains marginal, which is likely due to the efficiency of modern tensorized implementations. All in all, these results suggest that the main scalability bottleneck of the proposed framework lies in the quadratic growth with respect to the sample size, rather than in moderate increases in matrix dimension.

### B.2 Sensitivity to noise.

The proposed method assumes that the observed data can be well represented by an underlying matrix-manifold structure. Under severe noise or corruption, the estimated representations, such as Correlation matrices or Grassmannian subspaces, may become unstable, which can degrade the quality of the resulting low-dimensional embeddings. To empirically evaluate this effect, we extend the synthetic experiments in Section 5 by injecting noise with varying intensities. Fig. 17 shows that with the noise level increases, the separability between clusters corresponding to different categories gradually deteriorates. To further quantify the impact of noise, we adopt an evaluation protocol similar to de Surrel et al. (2025), where a high-dimensional geodesic is sampled and uniformly discretized into 50 points, which are then embedded into the low-dimensional space. We measure the deviation between the embedded trajectory and the corresponding ground-truth geodesic. Fig. 18 illustrates that higher noise levels lead to increasingly large deviations from the true geodesic. Moreover, the sensitivity to noise varies across different manifolds, although the overall degradation trend remains consistent. In particular, noticeable deviations occur at a noise level of 0.8 for the Grassmannian subspace, whereas Correlation and SPSD matrices are more sensitive, with evident degradation appearing around 0.3.

This discrepancy stems from their distinct algebraic constructions. The Grassmannian subspace acts as a geometric low-pass filter by retaining only the leading $q$ eigenvectors, making it inherently robust to isotropic additive noise. In contrast, the SPSD matrix preserves eigenvalue magnitudes, meaning additive noise directly inflates its trace and distorts quotient-geometric distances. Similarly, the Correlation matrix is highly sensitive because its diagonal normalization ($C = \Upsilon^{-1/2}\Sigma\Upsilon^{-1/2}$) disproportionately amplifies noise in low-variance dimensions, creating spurious correlations that are further exacerbated by the Cholesky decomposition in the PHC metric.

# C   Detailed table of results for different dimensionality-reduction algorithms across two datasets.

## C.1   Trustworthiness

Table 7: Experimental evaluation of Trustworthiness on CIFAR-10 and Tiny-ImageNet datasets.

| Dataset | Manifold | Method | Percent of Neighbors ($k$) | | | | |
|---|---|---|---|---|---|---|---|
| | | | 10% | 20% | 30% | 40% | 50% |
| **CIFAR-10** | Grassmann | Ours (Riem+Riem) | **0.8372** | **0.8445** | **0.8411** | **0.8270** | **0.7901** |
| | | Ours (Riem+Euc) | 0.8352 | 0.8314 | 0.7994 | 0.7389 | 0.6006 |
| | | t-SNE | 0.5365 | 0.5505 | 0.5600 | 0.5542 | 0.5039 |
| | | UMAP | 0.5411 | 0.5473 | 0.5566 | 0.5520 | 0.5005 |
| | | GDMaps (Riem+Riem) | 0.8255 | 0.8353 | 0.8332 | 0.8172 | 0.7824 |
| | | GDMaps (Riem+Euc) | 0.6407 | 0.6102 | 0.5901 | 0.5694 | 0.5190 |
| | Correlation | Ours (Riem+Riem) | **0.9500** | **0.9463** | **0.9323** | **0.9198** | **0.8999** |
| | | Ours (Riem+Euc) | 0.9258 | 0.8885 | 0.8452 | 0.8033 | 0.7560 |
| | | t-SNE | 0.8691 | 0.7959 | 0.7548 | 0.7189 | 0.6623 |
| | | UMAP | 0.9257 | 0.8680 | 0.8142 | 0.7545 | 0.6836 |
| | | MDS | 0.8530 | 0.8409 | 0.8320 | 0.8202 | 0.7905 |
| | SPSD | Ours (Riem+Riem) | **0.9376** | **0.9320** | **0.9222** | **0.9051** | **0.8804** |
| | | Ours (Riem+Euc) | 0.9352 | 0.9182 | 0.9085 | 0.8934 | 0.8637 |
| | | SPD-SNE (Riem+Riem) | 0.9243 | 0.9072 | 0.8830 | 0.8476 | 0.7968 |
| | | SPD-SNE (Riem+Euc) | 0.9066 | 0.8789 | 0.8541 | 0.8229 | 0.7717 |
| **Tiny-ImageNet** | Grassmann | Ours (Riem+Riem) | **0.7893** | **0.7685** | **0.7549** | **0.7384** | **0.6941** |
| | | Ours (Riem+Euc) | 0.7860 | 0.7614 | 0.7377 | 0.7070 | 0.6163 |
| | | t-SNE | 0.5263 | 0.5441 | 0.5552 | 0.5504 | 0.5016 |
| | | UMAP | 0.5263 | 0.5417 | 0.5544 | 0.5493 | 0.4991 |
| | | GDMaps (Riem+Riem) | 0.7825 | 0.7654 | 0.7528 | 0.7348 | 0.6905 |
| | | GDMaps (Riem+Euc) | 0.6279 | 0.5995 | 0.5872 | 0.5708 | 0.5154 |
| | Correlation | Ours (Riem+Riem) | **0.9324** | **0.9213** | **0.9157** | **0.9076** | **0.8864** |
| | | Ours (Riem+Euc) | 0.9096 | 0.8548 | 0.8493 | 0.8280 | 0.7918 |
| | | t-SNE | 0.8770 | 0.7561 | 0.7184 | 0.6974 | 0.6550 |
| | | UMAP | 0.9160 | 0.7435 | 0.7364 | 0.7067 | 0.6885 |
| | | MDS | 0.7852 | 0.7731 | 0.7674 | 0.7555 | 0.7200 |
| | SPSD | Ours (Riem+Riem) | **0.9392** | **0.8972** | **0.8725** | 0.8431 | 0.8046 |
| | | Ours (Riem+Euc) | 0.9382 | 0.8946 | 0.8698 | **0.8594** | **0.8449** |
| | | SPD-SNE (Riem+Riem) | 0.9278 | 0.8799 | 0.8439 | 0.8029 | 0.7435 |
| | | SPD-SNE (Riem+Euc) | 0.9112 | 0.8420 | 0.7930 | 0.7569 | 0.7069 |

## C.2 KNN Classification Accuracy

Table 8: Experimental evaluation of KNN classification accuracy on CIFAR-10 and Tiny-ImageNet datasets.

| Dataset | Manifold | Method | Number of Neighbors ($k$) | | | | |
|---|---|---|---|---|---|---|---|
| | | | 1 | 7 | 15 | 23 | 31 |
| CIFAR-10 | Grassmann | Raw Data | 0.604±0.025 | 0.727±0.029 | 0.771±0.025 | 0.804±0.019 | 0.806±0.024 |
| | | Ours | **0.720±0.022** | **0.766±0.027** | **0.769±0.036** | **0.777±0.034** | **0.776±0.030** |
| | | GDMaps | 0.449±0.022 | 0.444±0.028 | 0.436±0.024 | 0.423±0.014 | 0.433±0.010 |
| | | t-SNE | 0.236±0.040 | 0.250±0.034 | 0.233±0.051 | 0.234±0.034 | 0.212±0.042 |
| | | UMAP | 0.243±0.022 | 0.284±0.030 | 0.277±0.021 | 0.258±0.032 | 0.255±0.026 |
| | Correlation | Raw Data | 0.830±0.029 | 0.829±0.020 | 0.825±0.024 | 0.812±0.023 | 0.811±0.026 |
| | | Ours | **0.964±0.010** | **0.976±0.007** | **0.978±0.006** | **0.972±0.010** | **0.973±0.009** |
| | | MDS | 0.853±0.031 | 0.887±0.026 | 0.888±0.021 | 0.879±0.015 | 0.876±0.026 |
| | | t-SNE | 0.957±0.008 | 0.965±0.011 | 0.967±0.012 | 0.961±0.010 | 0.953±0.006 |
| | | UMAP | 0.956±0.014 | 0.960±0.008 | 0.962±0.010 | 0.964±0.013 | 0.964±0.010 |
| | SPSD | Raw Data | 0.810±0.026 | 0.870±0.030 | 0.888±0.021 | 0.887±0.011 | 0.887±0.022 |
| | | Ours | **0.905±0.012** | **0.934±0.007** | **0.936±0.008** | **0.936±0.011** | **0.936±0.011** |
| | | SPD-SNE | 0.898±0.006 | 0.922±0.015 | 0.914±0.018 | 0.910±0.016 | 0.905±0.017 |
| Tiny-ImageNet | Grassmann | Raw Data | 0.772±0.021 | 0.865±0.023 | 0.873±0.019 | 0.880±0.013 | 0.877±0.013 |
| | | Ours | **0.657±0.018** | **0.728±0.008** | **0.733±0.013** | **0.741±0.019** | **0.739±0.021** |
| | | GDMaps | 0.455±0.019 | 0.474±0.013 | 0.479±0.014 | 0.481±0.013 | 0.483±0.010 |
| | | t-SNE | 0.191±0.014 | 0.191±0.017 | 0.193±0.026 | 0.175±0.015 | 0.163±0.025 |
| | | UMAP | 0.236±0.032 | 0.243±0.018 | 0.236±0.021 | 0.216±0.029 | 0.206±0.028 |
| | Correlation | Raw Data | 0.952±0.016 | 0.965±0.012 | 0.962±0.020 | 0.957±0.018 | 0.945±0.017 |
| | | Ours | **0.999±0.001** | 0.998±0.001 | **0.998±0.001** | **0.998±0.001** | 0.997±0.002 |
| | | MDS | 0.804±0.028 | 0.850±0.020 | 0.842±0.029 | 0.835±0.031 | 0.818±0.030 |
| | | t-SNE | 0.998±0.002 | 0.997±0.002 | 0.997±0.002 | 0.997±0.002 | 0.997±0.002 |
| | | UMAP | 0.998±0.001 | **0.999±0.001** | 0.997±0.002 | 0.997±0.002 | **0.998±0.001** |
| | SPSD | Raw Data | 0.953±0.008 | 0.971±0.010 | 0.976±0.011 | 0.977±0.013 | 0.985±0.007 |
| | | Ours | **0.990±0.006** | **0.990±0.006** | **0.992±0.005** | **0.992±0.005** | **0.988±0.009** |
| | | SPD-SNE | 0.978±0.008 | 0.980±0.012 | 0.975±0.008 | 0.978±0.010 | 0.972±0.011 |

