# OpenReview forum: "Visualization of High-Dimensional Matrix Manifolds"
_TMLR — Decision pending for TMLR_

### Review · Reviewer_exvq · 2026-02-23

**Summary Of Contributions:**

This paper proposes a new way to visualize high-dimensional matrix data (Grassmann, Correlation, SPSD manifolds) by extending t-SNE to preserve curvature. It uses Riemannian metrics (e.g., geodesic distance) to maintain geometric structure, avoiding distortion from flat Euclidean space. It shows better performance than traditional methods (e.g., t-SNE) in preserving local clusters and class separation on benchmarks like MNIST and EEG data.

**Audience:**

Yes

**Audience Explanation:**

It focus on geometriy-aware visualization of matrix manifolds appeals to TMLR's audience interested in non-Euclidean data and dimensionality reduction.

**Broader Impact Concerns:**

There is no ethical issue.

**Claims And Evidence:**

Yes

**Claims Explanation:**

This paper proposes a new way to visualize high-dimensional matrix data (Grassmann, Correlation, SPSD manifolds) by extending t-SNE to preserve curvature. Clear experiments comparing results on real-world data.

**Requested Changes:**

1. The claim about “preserving curvature information” seems a bit strong; at least the authors did not provide a theorem about curvature preservation. At the same time, the method does not explicitly compute or constrain curvature as an independent quantity, but uses it implicitly through geodesic distances, probability construction, and manifold optimization. I am afraid that there may be a (minor) gap between the claim and what is actually implemented. So I suggest the authors soften the wording to better clarify this point.

2. It is suggested that the authors may consider adding experiments on larger-scale datasets to further demonstrate performance in more practical settings. The current experiments mainly focus on synthetic data, moderate-size benchmarks, and selected EEG features, which may not fully reflect how the method behaves on truly large-scale or high-dimensional real-world data.  If the authors believe the current experiments are sufficient, they could briefly explain why additional large-scale evaluation is not necessary.

3. It may be helpful to include a discussion on the computational complexity of the proposed method. Since pairwise geodesic distances and Riemannian optimization are used, it is not clear how the method scales with respect to the number of samples and matrix dimensions. A brief analysis or empirical runtime comparison would improve understanding of its practical applicability.

4. It may also be helpful to explain terms like “Riemannian geometry” in a simpler and more intuitive way for non-expert readers, for example by comparing curved and flat spaces. If the current level of detail is preferred, a short intuitive remark could still improve readability.

5. It is suggested to include a brief discussion on possible limitations of the method, such as performance under noisy data or unclear manifold structure. If this is not a concern in the current setting, the authors could clarify this point.

---

> ### Author Response · Authors · 2026-04-25
>
> We sincerely appreciate your constructive and inspiring comments. We address each one below.
>
> **Response to Comment 1**:
> We agree that the original wording may be overly strong, as our method does not explicitly estimate or constrain curvature as an independent quantity. In the revised version, we have softened the wording throughout the paper. Specifically, we replaced expressions such as “preserving curvature” with more suitable descriptions such as “geometry-aware” or “respecting intrinsic manifold geometry”.
>
> **Response to Comment 2:**
> According to your suggestion, we have added additional experiments on the CIFAR-10 and Tiny-ImageNet datasets. For each dataset, we randomly choose 10 classes, each containing 100 samples. Within each class, 50 images are first randomly selected and then vectorized. PCA is subsequently applied to obtain a $50 \times 32$ feature matrix. Based on it, each feature matrix is further transformed into three types of manifold-valued representations. Specifically, a Grassmannian representation given by an orthonormal basis matrix of size $32 \times 25$, and Correlation and SPSD representations, both represented as $32 \times 32$ matrices. The visualization results consistently show that compared with existing visualization methods, there is clearer class separation across all three manifold-valued data types on both datasets.
>
> This further confirms that the proposed framework generalizes beyond the datasets involved in the main paper. Please kindly refer to **Section 5.5, Fig. 14, and Fig. 15 (pages 17-19)** for detailed information.
>
> **Response to Comment 3:**
> In summary, the proposed method requires $N^2$ comparisons across $N$ samples to compute the pairwise Riemannian distances, where each evaluation involves nontrivial manifold-specific operations.
>
> For the Grassmann manifold, the distance computation relies on SVD, resulting in a complexity of $\mathcal{O}(N^2q^2 (d + q))$, where $N$, $d$, and $q$ signify the number of samples, the ambient space dimension, and the subspace dimension, respectively. For the Correlation and SPSD manifolds, the distance computations involve Cholesky and eigenvalue decompositions, resulting in the complexities of $\mathcal{O}(Nk^3 + N^2k^2)$ and $\mathcal{O}(N^2k^3)$, respectively. Wherein, $k$ denotes the number of rows and columns of the Correlation and SPSD matrices.
>
> As shown in Table A, we also provide an empirical runtime comparison under different matrix manifold-valued representations, matrix dimensions, and sample sizes to illustrate the practical computational cost.
>
> **Table A: Runtime comparison (s/iteration) under different matrix manifold-valued representations, sample sizes, and matrix size($n$ represents the sample sizes.).**
>
> | Data types | Matrix size | $n=100$ | $n=300$ | $n=500$ | $n=700$ | $n=1000$ |
> |:-:|:-:|:-:|:-:|:-:|:-:|:-:|
> | Correlation matrix | 4×4 | 0.040 | 0.129 | 0.207 | 0.294 | 0.565 |
> | | 8×8 | 0.040 | 0.124 | 0.215 | 0.294 | 0.549 |
> | | 16×16 | 0.041 | 0.138 | 0.198 | 0.328 | 0.533 |
> | | 32×32 | 0.041 | 0.136 | 0.210 | 0.315 | 0.528 |
> | SPSD matrix | 4×4 | 0.035 | 0.158 | 0.336 | 0.581 | 1.068 |
> | | 8×8 | 0.034 | 0.156 | 0.365 | 0.583 | 1.082 |
> | | 16×16 | 0.035 | 0.155 | 0.334 | 0.581 | 1.093 |
> | | 32×32 | 0.034 | 0.158 | 0.345 | 0.578 | 1.079 |
> | Grassmannian subspace | 4×2 | 0.097 | 0.452 | 1.074 | 1.857 | 3.507 |
> | | 8×4 | 0.103 | 0.476 | 1.128 | 1.930 | 3.693 |
> | | 16×8 | 0.104 | 0.483 | 1.213 | 1.881 | 3.531 |
> | | 32×16 | 0.106 | 0.477 | 0.998 | 1.846 | 3.545 |
>
> Please kindly refer to **Appendix B.1, Fig. 17, and Table 4 in pages 28-30** of the revised manuscript for detailed information.
>
> **Response to Comment 4**:
> To improve the readability of the manuscript for non-expert readers, we have revised the preliminary section to provide a clearer explanation of the distinctions and connections between curved and flat spaces. Specifically, we have added the following Table B to compare the fundamental operations defined in Euclidean space and Riemannian manifolds. Furthermore, an illustrative figure has been included to offer a more intuitive understanding. For detailed information, please kindly refer to the **Riemannian geometry part**, **Table 1, and Fig. 2 in Section 3 (pages 4, 5)** of the revised manuscript.
>
> **Table B: Comparison of basic operations in Euclidean space and Riemannian manifolds**
> |Operation|Euclidean space|Riemannian manifold|
> |:-:|:-:|:-:|
> |Straight line|Straight line|Geodesic|
> |Subtraction|$\overrightarrow{xy}=y-x$|$\overrightarrow{xy}=\mathrm{Log}_x(y)$|
> |Addition|$y=x+\overrightarrow{xy}$|$y=\mathrm{Exp}_x(\overrightarrow{xy})$|
> |Parallelly moving|$v\to v$|$\mathrm{PT}_{x \to y}(v)$|

---

> > ### Author Response · Authors · 2026-04-25
> >
> > **Response to Comment 5**:
> > Under noisy conditions, the estimated manifold representations (e.g., SPSD matrices or subspaces) may become unreliable, thereby degrading the quality of the resulting visualizations.
> >
> > To further substantiate this observation, we embed discretized geodesic trajectories and evaluate their deviations from the corresponding ground-truth trajectories. As shown in **Fig. 19 of the revised manuscript** (**Section B.2, page 30**), increasing noise consistently degrades performance across all three manifold-valued representations. In particular, noticeable deviations emerge at a noise level of 0.8 for the Grassmannian subspace, whereas the Correlation and SPSD matrices are more sensitive, with evident degradation appearing around 0.3.
> >
> > This difference can be attributed to their distinct algebraic constructions and geometric properties. The Grassmannian subspace effectively behaves as a geometric low-pass filter: during construction, it retains only the leading $q$ eigenvectors while discarding the associated eigenvalue magnitudes, which makes it inherently robust to isotropic additive noise. By contrast, the SPSD matrix preserves eigenvalue magnitudes. As a result, additive noise directly perturbs the spectral energy and inflates the matrix trace, shifting samples along the positive semi-definite cone and causing immediate distortions under the quotient-geometric metric. The Correlation matrix is also sensitive to noise due to the diagonal normalization step ($C=\Upsilon^{-1/2}\Sigma\Upsilon^{-1/2}$). In this case, noise can disproportionately affect dimensions with low intrinsic variance, thereby inducing spurious correlations. These perturbations are further amplified by the Cholesky decomposition involved in the PHC metric, which leads to a faster degradation of geodesic trajectories at relatively low noise levels.

---

> ### Comment · Reviewer_exvq · 2026-04-26
>
> Many thanks for the response and revisions. The added experiments on CIFAR-10 and Tiny-ImageNet datasets are encouraging. I wonder if it is possible for the authors to open-source the implementation such that the readers can better benefit from your method.
>
> I also suggest the authors proofread the manuscript more carefully to make the reference format more consistent. For example, in the following bib item, "the Journal of machine Learning research" shoud be "The Journal of Machine Learning Research".
>
> > Fabian Pedregosa, Gaël Varoquaux, Alexandre Gramfort, Vincent Michel, Bertrand Thirion, Olivier Grisel, Mathieu Blondel, Peter Prettenhofer, Ron Weiss, Vincent Dubourg, et al. Scikit-learn: Machine learning in python. the Journal of machine Learning research, 12:2825–2830, 2011.

---

> ### Author Response · Authors · 2026-05-04
>
> Dear $\textit{Reviewer exvq}$
>
> We sincerely thank you for the positive feedback, encouragement, and valuable suggestions.
>
> **Regarding open-sourcing the implementation**:
>
> We fully agree that making the implementation publicly available would benefit the readers and research community. To comply with the double-blind review policy of TMLR, we have provided the main source codes through the following anonymous repository:
>
> https://anonymous.4open.science/r/ManiReduce-8D67/
>
> Upon acceptance, we will release the complete, well-documented codebase in a public GitHub repository, including all implementation details, scripts, and instructions needed to reproduce the experimental results.
>
> **Regarding reference formatting**:
>
> We appreciate you for pointing out this inconsistency. We have thoroughly proofread the manuscript and carefully revised all bibliography entries to ensure consistent formatting and proper capitalization throughout. Since we are still addressing the remaining comments from other reviewers, we will incorporate the reference-formatting corrections together with the other revisions and upload the revised manuscript upon completion.
>
> Thank you again for taking the time to review our manuscript and for your helpful and constructive comments.
>
> Best regards
>
> Authors of the Submission #6937

---

> > ### Comment · Reviewer_exvq · 2026-05-04
> >
> > Thanks for the feedback. I have no further questions.

---

### Review · Reviewer_FGpc · 2026-03-08

**Summary Of Contributions:**

The authors propose a Riemannian generalization of t-SNE specifically designed for high-dimensional data lying on matrix manifolds, addressing the key limitation of standard Euclidean visualization tools (including vanilla t-SNE), which distort the intrinsic curvature and non-Euclidean geometry of such data.

**Audience:**

Yes

**Audience Explanation:**

Researchers working on geometric deep learning, Riemannian optimization, and structured data representations could use the findings of this paper.

The paper's practical angle, better visualizations for EEG, image sets, and covariance-based features appeal to applied ML folks in signal processing and computer vision who use these representations.

**Claims And Evidence:**

Yes

**Claims Explanation:**

- Math and method are solid: detailed derivations, geodesic distances, Riemannian gradients, and optimization steps are clearly explained and look correct.

- Experiments feel convincing: synthetic + real data (MNIST subspaces, EEG), multiple baselines, metric comparisons, and nice visualizations showing better cluster separation and geometry preservation.

- Writing is clear, logical, and reproducible in principle — no obvious overclaims or weak spots.

**Requested Changes:**

Minor changes:
- I suggest moving the code to the Appendix and providing a more visual demo of your proposed method in Figure 1, rather than putting Python code.

---

> ### Author Response · Authors · 2026-04-25
>
> We sincerely thank you for your careful review of our submission and for your encouraging feedback. Following your suggestion, we have moved the code to the Appendix (**Fig. 16, page 24**). In addition, we have redesigned Fig. 1 to provide a clearer visual overview of the proposed method. The new **Fig. 1 (page 2)** illustrates the process of inputting high-dimensional data into our method to obtain low-dimensional embeddings, including converting the input data into a manifold-valued representation, applying Riemannian metrics for similarity computation, and generating the final low-dimensional visualization using our Riemannian t-SNE. For detailed information, please kindly refer to the revised manuscript.

---

### Review · Reviewer_4zwL · 2026-04-25

**Summary Of Contributions:**

The paper presents dimensionality reduction and visualization framework that adapts the popular t-SNE algorithm for high-dimensional matrix manifolds. The overall idea is to replaces standard Gaussian and Student t-distributions with a Riemannian normal distribution for high-dimensional spaces and a heavy-tailed t-distribution defined by geodesic distance for low-dimensional spaces. However, this exact mathematical paradigm has already been explored by prior works for specific (matrix) manifolds (Guo et al. 2022; de Surrel et al. 2025).

**Audience:**

Yes

**Audience Explanation:**

Yes. The practitioners working on the three matrix-manifold valued data may find the implementations useful.

**Broader Impact Concerns:**

There are no significant ethical implications or broader impact concerns.

**Claims And Evidence:**

No

**Claims Explanation:**

- The submission presents the proposed Riemannian t-SNE framework as a core contribution. However, this framework has already been proposed and employed in existing works (Guo et al. 2022; de Surrel et al. 2025). While the paper explicitly mentions that its contribution is restriction to three matrix manifolds, it is unclear whether this was a trivial extension of prior works or whether some technical/empirical/theoretical challenge existed for the three considered matrix manifolds

- For the SPSD manifold, the paper compare the proposed method against SPD-SNE. The propsoed method operates on low-rank matrices while SPD-SNE enforces full-rank positive definite constraints. The claim of superior local neighborhood preservation over a baseline that was forced to process mismatched data does not constitute convincing evidence of algorithmic superiority.

**Requested Changes:**

- The paper should explicitly acknowledge that Eqs (9)-(11) have been explored in prior Riemannian t-SNE works for various manifolds.
- In Section 4.4, the manifold agnostic expressions of Riemannian gradient, retraction, etc. are provided. For the three considered manifolds, prior manifold optimization works (such as those by P.-A. Absil and others) would have discussed manifold specific expressions. The paper should acknowledge them as well with context.
- The paper should address the SPSD evaluation issue discussed previously
- The paper should discuss the computational complexity of their algorithm on the three considered manifolds
- Finally, the paper should articulate their contributions in the light of above changes

---

> ### Author Response · Authors · 2026-05-20
>
> Thank you very much for your constructive comments. We address each one below:
>
> ### Response to Comment 1
>
> We fully agree that the general mechanism used in Eqs. (9)–(11), constructing probability distributions from geodesic distances and minimizing a KL divergence in the spirit of t-SNE, has been explored in prior Riemannian extensions such as CO-SNE (Guo et al., 2022) and SPD-SNE (de Surrel et al., 2025). Following your suggestion, we have revised the manuscript to explicitly acknowledge that these equations follow the established Riemannian t-SNE paradigm rather than constitute a novel objective (Sections 4, page 6 and Sections 4.3, page 9).
>
> We have also clarified the motivation for extending this mechanism to the three matrix manifolds considered in this paper. **These manifolds are not chosen arbitrarily, but correspond to three common and complementary forms of structured matrix-valued data in machine learning.** Specifically, Grassmann manifolds naturally represent subspace-valued data, Correlation manifolds encode scale-invariant dependency structures, and fixed-rank SPSD manifolds model low-rank or degenerate covariance structures. Such data frequently arises in applications including image set classification [Wang et al., IEEE-TCDS 2022, pp: 957-970; Wang et al., IEEE-TMM 2021, pp: 228-242; Wei et al., IEEE-TMM 2022, pp: 4307-4322], EEG decoding [Hu et al., IJCAI 2025, pp: 5372-5380; Wang et al., IJCAI 2024, pp: 5099-5107; Wang et al., NeurIPS, 2025, pp: 112051-112091], and action recognition [Chen et al., ICLR 2024; Nguyen & Yang, ICML 2023, pp: 26031-26062]. Therefore, developing geometry-aware visualization methods for these manifolds is practically important, particularly because conventional Euclidean t-SNE may distort their intrinsic geometric structure.
>
> **The key motivation of our work is that existing Riemannian t-SNE methods do not directly cover these three settings.** CO-SNE is designed for the Poincaré ball with constant negative curvature, whereas Grassmann manifolds exhibit fundamentally different curvature properties. SPD-SNE focuses on full-rank SPD matrices, while fixed-rank SPSD manifolds require a quotient-geometric formulation to preserve rank constraints. Correlation manifolds, in contrast, impose unit-diagonal constraints and are equipped in our work with a PHC-induced geometry via a diffeomorphic mapping to a hyperbolic product space. These geometric characteristics make a direct application of prior hyperbolic or SPD-based t-SNE formulations insufficient.
>
> **Accordingly, our contribution does not lie in the generic KL-based Riemannian t-SNE objective itself, but in the principled extension of this mechanism to three geometrically distinct and practically important matrix manifolds.** This extension requires manifold-specific definitions of geodesic distances, Riemannian gradients, and retraction/update operations, ensuring that the learned low-dimensional embeddings remain on the appropriate target manifolds throughout optimization. In the revised manuscript, we have added a new subsection (Section 4.5, page 10) to detailedly discuss both the connections to and distinctions from existing Riemannian t-SNE approaches. We have also clarified that the main contribution of this work lies in extending the established Riemannian t-SNE paradigm to several practically important matrix manifolds, thereby enriching the methodological toolbox for manifold-valued dimensionality reduction and visualization.
>
> ### Response to Comment 2
>
> We agree that the manifold-specific optimization components in Section 4.4 (page 9), including Riemannian gradients, retractions, and constraint-preserving updates, should be more clearly contextualized within the existing literature on manifold optimization.
>
> Following your suggestion, we have revised Section 4.4 to better acknowledge the relevant theoretical foundations. In particular, we now explicitly reference the general Riemannian optimization framework, including Absil et al. [a] and Boumal et al. [b], as well as classical results on Grassmannian geometry and optimization, such as Edelman et al. [c].
>
> In addition, we clarify that our contribution is not to propose new optimization operators, but rather to integrate appropriate manifold-specific optimization tools into a unified Riemannian t-SNE framework for visualization.
>
> [a] P-A Absil, Robert Mahony, and Rodolphe Sepulchre. Optimization algorithms on matrix manifolds. Princeton University Press, 2008.
>
> [b] Nicolas Boumal. An introduction to optimization on smooth manifolds. Cambridge University Press, 2023.
>
> [c] Alan Edelman, Tomás A Arias, and Steven T Smith. The geometry of algorithms with orthogonality constraints. SIAM Journal on Matrix Analysis and Applications, 20(2):303–353, 1998.

---

> ### Author Response · Authors · 2026-05-20
>
> ### Response to Comment 3
> We thank you for raising this important concern regarding the SPSD evaluation and the fairness of comparing our method with SPD-SNE. We answer this question from the following three aspects:
>
> **On the choice of SPD-SNE as a baseline.** We agree that the comparison is inherently asymmetric, as SPD-SNE is designed for full-rank SPD matrices, whereas our method operates on SPSD matrices. However, to the best of our knowledge, there currently exists no visualization method specifically designed for low-rank SPSD manifolds. We therefore used SPD-SNE only as the closest available surrogate baseline, rather than as a fully matched counterpart.
>
> Importantly, the purpose of this comparison is **not to claim algorithmic superiority under matched assumptions**, but rather to illustrate what happens when inherently low-rank or near-rank-deficient data are forced into a full-rank SPD geometry, and thereby motivating the need for an SPSD-aware visualization method.
>
>
> **The necessity of the SPSD method in practice.** In many real-world settings, covariance matrices are naturally rank-deficient. For example, when the number of observations $T$ is smaller than the feature dimension $d$, the sample covariance $\Sigma = \frac{1}{T-1}\bar{X}\bar{X}^\top$ is rank-deficient by construction and lies on the boundary of the SPD cone.
>
> A common workaround is to enforce positive definiteness via regularization (e.g., $\Sigma + \varepsilon I$). However, this operation can artificially inflate very small eigenvalues, alter the intrinsic low-rank geometry, and remove meaningful rank-deficient structure from the data. In contrast, the SPSD manifold explicitly models such low-rank structures and preserves the corresponding quotient geometry.
>
> **Experiment under ill-conditioned regimes.** To better address your concern, we have added analysis on CIFAR-10 and Tiny-ImageNet (Section 5.5, page 20). Specifically, for each sample, we first randomly select 50 images from the same class and vectorize them. PCA is then applied to obtain a feature matrix of size (50 $\times$ 32), which captures the dominant feature of the data. We then compute the corresponding covariance matrix and construct the SPSD representation by truncated eigendecomposition. This setting naturally produces ill-conditioned or near-rank-deficient covariance matrices, which is the intended regime of the proposed SPSD-aware method.
>
> We quantify this property using the condition number:
>
> $$\kappa(A) = \frac{\sigma_{\max}(A)}{\sigma_{\min}(A)},$$
>
> where $\sigma$ represents the singular value of the SPSD matrix $A$. As summarized in Table A, a substantial proportion of matrices are highly ill-conditioned:
>
> **Table A: Proportion of ill-conditioned matrices**
> |Dataset|$\kappa > 5\times10^2$|$\kappa > 1\times10^3$|$\kappa > 1.5\times10^3$|
> |:-:|:-:|:-:|:-:|
> |CIFAR-10|100.00%|91.30%|62.10%|
> |Tiny-ImageNet|100.00%|90.80%|29.70%|
>
> Consistent with this observation, the visualization results (Figs. 14(e,f) and 15(e,f)) show that our method yields clearer cluster separation compared to SPD-SNE in these ill-conditioned scenarios. We have clarified in the revised manuscript that this result should be interpreted as evidence for the suitability of the proposed method for low-rank/near-rank-deficient SPSD data, rather than as a direct claim of superiority over SPD-SNE under its full-rank assumptions.

---

> ### Author Response · Authors · 2026-05-20
>
> ### Response to Comment 4
>
> The computational complexity of the proposed algorithm across the three considered matrix manifolds is discussed in **Appendix B.1** (pages 28–30) of the revised manuscript. For clarity, we summarize the main results below.
>
> The dominant computational cost arises from the pairwise Riemannian distance computations among $N$ samples. For the **Grassmann manifold**, computing all pairwise distances involves repeated SVD-based operations, resulting in a complexity of $O(N^2 q^2 (d + q))$, where $N$, $d$, and $q$ denote the number of samples, the ambient dimension, and the subspace dimension, respectively. For the **Correlation manifold**, the PHC-based distance computation consists of two stages: a one-time Cholesky decomposition for all $N$ samples with cost $O(Nk^3)$, followed by pairwise distance evaluations in the associated hyperbolic product space with cost $O(N^2 k^2)$. This yields an overall complexity of $O(Nk^3 + N^2 k^2)$, where $k$ denotes the number of rows and columns of the matrix. For the **SPSD manifold**, the quotient-geometric distance requires pairwise matrix square-root operations and eigenvalue decompositions, leading to a complexity of $O(N^2 k^3)$.
>
> We further support this theoretical analysis with empirical runtime measurements reported in Table 5 and Figure 17 (Appendix B.1, page 28). The results show that the runtime increases rapidly with the number of samples $N$, which is consistent with the quadratic dependence induced by pairwise distance computations. Among the three settings, the Grassmann manifold incurs the highest computational cost, primarily due to repeated SVD operations and tangent space projections during Riemannian optimization. In contrast, the Correlation and SPSD manifolds are relatively more efficient in our experiments.
>
> Overall, the primary scalability bottleneck of the proposed framework lies in its quadratic dependence on the number of samples, whereas moderate increases in matrix size have a comparatively smaller impact, as observed in our empirical evaluation.
>
> ### Response to Comment 5
> **Summary of contributions in light of the above changes.**
> In response to the above comments, we have revised the contribution statements in Section 1 (page 3) to more accurately position our work with respect to prior Riemannian t-SNE and manifold optimization literatures. We now explicitly acknowledge that the general Riemannian t-SNE mechanism, as well as standard Riemannian optimization-related operators, are established tools rather than new contributions of this paper.
>
> Accordingly, our contributions are now characterized as follows:
>
> (1) **Manifold-specific instantiation of Riemannian t-SNE.**
> Building upon the established Riemannian t-SNE paradigm, we instantiate a geometry-aware visualization framework on three practically important matrix manifolds: Grassmann, Correlation, and fixed-rank SPSD manifolds. The key contribution of our work lies in adapting the required geodesic distances, Riemannian gradients, and retraction/update operations to these distinct geometric settings.
>
> (2) **Geometry-aware visualization on the Grassmann manifold.**
> For subspace-valued data, we incorporate multiple Grassmannian metrics to study how different geometric distances impact the resulting visualization. This provides a more geometry-consistent alternative to Euclidean visualization methods and hyperbolic embeddings that may suffer from curvature mismatch.
>
> (3) **Visualization on the underexplored Correlation and SPSD manifolds.**
> For Correlation and fixed-rank SPSD matrices, we develop manifold-aware visualization strategies that respect their intrinsic constraints, including unit-diagonal correlation structure and low-rank positive semi-definite geometry. We have also clarified that the SPSD-related comparative experiments are intended to demonstrate the suitability of the proposed SPSD-aware method for low-rank or near-rank-deficient data, rather than to claim its direct superiority over SPD-SNE under mismatched assumptions.
>
> (4) **Experimental validity and computational transparency.**
> We evaluate the proposed framework on synthetic data, image sets, EEG signals, and network features. In addition, we have added experiments on CIFAR-10 and Tiny-ImageNet to further assess its scalability and generalization on more challenging visual benchmarks. We also provide theoretical and empirical computational complexity analyses for all three matrix manifolds. These experiments support the practical usefulness of the proposed manifold-specific visualization framework while clarifying where the computational overhead comes from.

---

> ### Author Response · Authors · 2026-06-25
>
> Dear Reviewer 4zwL,
>
> I hope this message finds you well.
>
> Thank you very much for your valuable comments and suggestions on our submission. We have carefully addressed your concerns in our response and sincerely appreciate the time and effort you have devoted to reviewing our work.
>
> We are writing to kindly follow up on the rebuttal discussion. We would like to know whether you have any additional questions or suggestions regarding our clarifications. We would be happy to provide any further explanations if needed.
>
> Thank you again for your time and consideration. Looking forward to hearing from you.
>
> Best regards,
>
> Authors of TMLR submission #6937

---

### Review · Reviewer_hsL1 · 2026-05-04

**Summary Of Contributions:**

The paper studies information visualization of data that lies on pre-defined manifolds, by changing the distance measure in SNE-like objective with one that computes the distances along the manifold. Three concrete methods are proposed, for three different manifolds that have known analytic exponential and logarithm maps and hence are computationally efficient. The t-SNE objective is used as is and the optimization is done using Riemannian gradient descent. The three method variants are demonstrated in a series of empirical evaluations.

**Additional Comments:**

Apologies for submitting the review late. I considered directly the version that has already been adapted based on other reviewer's comments, but I avoided checking any of the reviews.

**Audience:**

Yes

**Audience Explanation:**

Even though the scientific novelty of the work is rather limited, with known analytic Riemannian distances plugged into the standard t-SNE objective without even verifying how the theoretical properties of t-SNE might changed, the concrete details for how to visualize data on the three specific manifolds are valuable and informative. The manifolds are relevant for many practitioners and the authors demonstrate visually and qualitatively that the method works reliably enough. With public open implementation, I can well see some practitioners willing to use the visualization methods, and there is some potential for follow-up work that would better analyse and evaluate the methods, or consider extensions for other manifolds.

**Claims And Evidence:**

No

**Claims Explanation:**

The paper makes three concrete contributions of claims: the framework that allows incorporating Riemannian distances into t-SNE, the concrete instantiations of the framework for the three manifolds considered, and comprehensive empirical evaluation. No theoretical claims are made, other than calling the approach 'principled'. No technical properties, e.g. the convergence speed or accuracy of the Riemannian gradient method, are evaluated and there are not claims relating to those. This is not a major problem, but it would have been nice to see something about how the algorithm works from a technical perspective. Is the Riemannian gradient descent necessary, or would starndard GD work almost as well?

The empirical evaluation is problematic in two respects. First, the claims are very vague. The authors state "Our method yields more interpretable and informative visualizations compared to existing methods", without explicating how interpretability is measured, and for the empirical evaluations no clear goals are defined; it is simply said the method "is evaluated across ... data sets". I would have expected to see concrete claims explaining e.g. specific ways how Euclidean methods are expected to fail in certain scenarios and then direct evaluation of that, but in practice most of the evidence appears to be verbal characterization of the visualizations.

Second, the empirical evaluation is missing details that make evaluation of even the undirected experimentation difficult. Figure 13 shows the only quantitative evaluation using trustworthiness as the only metric, but the authors do not even define the metric. Section 5.4 is supposed to explain it, but does not give an equation or even a reference. This is not just a matter of presentation, but it would be important to see how the performance metric accounts for the distance measure. The standard trustworthiness metric requires finding neighborhoods in some distance and if the authors here use the specific distance the method is using to form the embedding, then the result is limited to showing the method works as it should in a technical sense. In other words, it is an important experiment to have, but not yet validating the quality of the visualization with an independent criterion.

Overall, the three method variants are sensible and appear to work well, but the paper is not a particularly good fit for TMLR that emphasises careful validation of the claims as main evaluation criterion. Evaluation of visualization tools is always difficult, but it is still possible to go quite a bit further than what was done here; as purely subjective evaluation this paper falls a bit short of also the lower evaluation standards of visualization papers in generic machine learning conferences.

**Requested Changes:**

It is absolutely critical to formally define the trustworthiness metric, provide appropriate citations for it, and discuss in detail how the new distance measures are used in this evaluation phase. Now it remains unclear how the neighborhoods are even defined when computing this metric, and assuming you use the Riemannian metric here you would need to clearly explain what the reader should conclude from the slightly better trustworthiness compared to the baselines.

The overall evaluation is very qualitative and I know it will be difficult to improve that, but at least you should carefully check the claims that are being made about the performance and make sure you address them in the empirical part. If you make claims about interpretability then you should also evaluate interpretability, which typically would mean user experiments. I doubt this is what you want to do, so it would be easier to re-consider the claims. If the only thing you can show is "visually match better the generative process used when constructing the synthetic data" then it is better to state that directly.

Finally, the related work is a bit scarce, omitting e.g. some early works on Riemannian geometry in information visualization. For example, Peltonen et al. "Improved learning of Riemannian metrics for exploratory analysis" (2004) and Brand "Charting a manifold" (2003) considered some forms of local curvature in visualization tasks roughly around the same time as PGA was introduced. While there has been a surge of Riemannian methods since 2022 or so (CO-SNE etc), it is a good idea to cover also the early literature. It would also be a good idea to discuss briefly in which metrics the exponential and logarithmic maps can be computed as easily as in your work -- are there other manifolds you could still extend to, and what could be done if working on a manifold with no closed-form expressions for the distance?

---

> ### Author Response · Authors · 2026-06-02
>
> We sincerely thank the reviewer for your very constructive and inspiring comments. Your feedback has significantly improved the quality of our work. Below, we address your concerns point by point.
>
> ---
> ### 1. Response to Comment 1：Trustworthiness
>
> According to your suggestion, we have substantially expanded Section 5.5 (page 19), especially Section 5.5.1, to make the evaluation protocol explicit in the revised manuscript.
>
> **(1) Formal definition.** Following Venna and Kaski (2006), we now provide the complete mathematical definition of trustworthiness in Eq. 16 (page 20):
>
> $$
> T(k) = 1 - \frac{2}{Nk(2N-3k-1)} \sum_{i=1}^{N} \sum_{j \in \mathcal{U}_k(i)} \left(r(i,j) - k\right),
> $$
>
> where $N$ is the number of data points, $r(i,j)$ is the rank of point $j$ with respect to point $i$ in the original high-dimensional space, and $\mathcal{U}_k(i)$ denotes the set of points that are among the $k$-nearest neighbors of $i$ in the low-dimensional embedding space. The score lies in $[0,1]$, with larger values indicating fewer false neighbors and better local neighborhood preservation.
>
> **(2) Distance measurement protocol.** We also introduce a dedicated paragraph in Section 5.5.1 to clarify how distances are used when computing neighborhoods. For all methods, the high-dimensional reference neighborhoods are computed using the intrinsic Riemannian distance of the corresponding data manifold. This choice reflects the goal of evaluating whether an embedding preserves the intrinsic neighborhood structure of manifold-valued data.
>
> For the low-dimensional embeddings, we distinguish between the following cases:
>
> * For Euclidean baselines, including t-SNE, UMAP, and MDS, low-dimensional neighborhoods are computed using standard Euclidean distance, which is consistent with their flat embedding spaces.
> * For our Riemannian embeddings, we report two complementary protocols:
>
>   * **Intrinsic evaluation (Riem+Riem):** low-dimensional neighborhoods are computed using the corresponding Riemannian distance on the target manifold. This protocol evaluates the preservation of intrinsic manifold neighborhoods under the geometry assumed by the representation.
>   * **Extrinsic evaluation (Riem+Euc):** low-dimensional neighborhoods are computed using Euclidean distance on the embedding coordinates. This protocol is included as a non-circular sanity check, since it does not use the same Riemannian distance in the low-dimensional evaluation. It tests whether the learned representations remain meaningful even if the downstream analysis ignores the manifold metric.
>
> In particular, the result of **Riem+Riem** protocol should be interpreted as intrinsic-neighborhood preservation, while that of **Riem+Euc** protocol provides an additional robustness check under a conventional Euclidean evaluation.
>
> **(3) Result.** As shown in Figure 13 (page 20), our method performs favorably under both protocols. The improvement made by the **Riem+Riem** protocol indicates better preservation of the intrinsic manifold neighborhoods, while the competitive performance made by the **Riem+Euc** protocol suggests that the gains are not solely an artifact of evaluating with the same distance used in the manifold-aware objective.

---

> > ### Author Response · Authors · 2026-06-02
> >
> > ### 2. Response to Comment 2: Claims and Evaluation
> >
> > According to your suggestion, we have made two main changes in the revised manuscript. For your convenience, the changes are listed below:
> >
> > **（1）Revising and narrowing the claims.**
> > We have removed claims about improved interpretability unless they are directly supported by evaluation. In particular, the Contributions section (page 3) no longer claims that our method produces “more interpretable” visualizations. Instead, we now state more specific and verifiable claims tied to the geometry-preservation objectives of the paper.
> >
> > **（2）Expanding quantitative evaluation.**
> > To reduce reliance on qualitative visual inspection, we have substantially expanded Section 5.5 and added three complementary quantitative evaluations: trustworthiness, KNN classification accuracy, and Spearman's rank correlation coefficient. The explanations for these three quantitative evaluations are as follows:
> >
> > * **Trustworthiness** evaluates local neighborhood preservation. As explained in Comment 1, we now formally define the metric and explicitly specify the distance protocols used in both the original and embedded spaces.
> > * We add **KNN classification accuracy** as a label-based downstream evaluation. Since class labels are not used during dimensionality reduction, this metric evaluates whether the learned embeddings preserve class-relevant neighborhood structure independently of the optimization objective.
> > * We add **Spearman's rank correlation coefficient** to evaluate global structure preservation. While trustworthiness and KNN focus primarily on local neighborhoods and class-label consistency, Spearman's correlation coefficient measures whether the embedding preserves the global ordering of pairwise intrinsic distances. This directly addresses your request for a clearer explanation of how Euclidean methods are expected to fail: Euclidean distances may be reasonable local approximations on a smooth manifold, but they can become inaccurate proxies for intrinsic geodesic distances at larger scales, especially on curved manifolds.
> >
> > The new results provide a more concrete and claim-aligned evaluation. Brief results are shown in Tables A, B, and C. For detailed information, please refer to Figs. 13-14 and Table 3 (pages 20 and 22) in the main paper.
> >
> > **Table A: Brief results of trustworthiness evaluation on CIAFR-10 dataset (Grassmann manifold), where the percentage represents the proportion of neighbors considered.**
> > | Method | 10% | 20% | 30% | 40% | 50% |
> > |---|---:|---:|---:|---:|---:|
> > | **Ours** | **0.8372** | **0.8445** | **0.8411** | **0.8270** | **0.7901** |
> > | GDMaps | 0.8255 | 0.8353 | 0.8332 | 0.8172 | 0.7824 |
> > | t-SNE | 0.5365 | 0.5505 | 0.5600 | 0.5542 | 0.5039 |
> > | UMAP | 0.5411 | 0.5473 | 0.5566 | 0.5520 | 0.5005 |
> >
> > **Table B: Brief results of KNN evaluation on CIAFR-10 dataset (Grassmann manifold)，where $k$ represents the number of neighbors considered when evaluating each low-dimensional embedding point.**
> > | Method | k=1 | k=7 | k=15 | k=23 | k=31 |
> > |-|:-:|:-:|:-:|:-:|:-:|
> > | **Ours** | **0.720±0.022** | **0.766±0.027** | **0.769±0.036** | **0.777±0.034** | **0.776±0.030** |
> > | GDMaps | 0.449±0.022 | 0.444±0.028 | 0.436±0.024 | 0.423±0.014 | 0.433±0.010 |
> > | t-SNE | 0.236±0.040 | 0.250±0.034 | 0.233±0.051 | 0.234±0.034 | 0.212±0.042 |
> > | UMAP | 0.243±0.022 | 0.284±0.030 | 0.277±0.021 | 0.258±0.032 | 0.255±0.026 |
> >
> > **Table C: Brief results of the evaluation of Spearman’s rank correlation coefficient ($\rho_s$) on CIAFR-10 dataset (Grassmann manifold).**
> > | Metric | **Ours** | GDMaps | t-SNE | UMAP |
> > |:-:|:-:|:-:|:-:|:-:|
> > | $\rho_s$ | **0.6081** | 0.6047 | 0.0148 | 0.0044 |
> >
> > Taking the Grassmann manifold as an example, t-SNE and UMAP obtain a near-zero Spearman's correlation coefficient on CIFAR-10, whereas our method achieves substantially higher rank correlation. This suggests that the Euclidean baselines may fail to preserve the globally inherent distance order in this setting, while the proposed Riemannian framework better maintains global geometric relationships. The KNN results further demonstrate that the low-dimensional embeddings produced by our method preserve class-relevant neighborhood structure effectively across the evaluated manifolds and datasets.

---

> ### Author Response · Authors · 2026-06-02
>
> ### 3. Response to Comment 3: Missing early Riemannian visualization literature
>
> We sincerely thank you for pointing out these foundational works. We fully agree that acknowledging early explorations of local curvature and Riemannian metrics in visualization is important for providing a complete historical context.
>
> In the revised manuscript, we have expanded the "Riemannian DR methods" paragraph in Section 2 (Related Work, page 4) and added both Brand (2003) [a] and Peltonen et al. (2004) [b] to the reference list. We now discuss these works as early attempts to incorporate manifold structure, local curvature, or learned Riemannian metrics into exploratory visualization.
>
> **References**
> >[a] Charting a manifold.
>
> >[b] Improved learning of riemannian metrics for exploratory analysis.
>
> ---
> ### 4. Response to Comment 4: Extensions to other manifolds and manifolds without closed-form geometry
>
> We thank you for this helpful suggestion and agree that the scope and limitations of the proposed framework should be clarified. Therefore, we have added a summary table (Table D) that compares representative manifold families according to the key ingredients required by our framework.
>
> **Table D. Applying the dimensionality reduction framework to various manifolds**
> | Manifold family | Intrinsic distance | Exp/Log or retraction | Differentiable / backprop. | Suitability for our framework |
> | :--- | :--- | :--- | :--- | :--- |
> | Hyperbolic space | Closed-form | Closed-form | Yes | High |
> | Grassmann manifold | Closed-form | Closed-form | Yes | High |
> | Full-rank SPD manifold | Closed-form under common metrics | Closed-form under AIRM / LEM | Yes | High |
> | Correlation manifold | Tractable via PHC representation | Tractable via PHC representation | Yes | High |
> | Low-rank SPSD manifold | Tractable under quotient geometry | Tractable under quotient geometry | Yes | High |
> | Implicit manifolds | Approximate | Approximate | Possible but costly | Low--Medium |
>
> This table is intended to clarify when the proposed framework can be extended efficiently, rather than to claim that all listed manifolds have been implemented in this work. In general, our framework is very suitable when the original data manifold admits a tractable intrinsic distance for constructing high-dimensional affinities, and when the target embedding manifold supports efficient distance computation, gradient evaluation, and valid manifold updates through exponential maps or retractions. The three manifolds studied in this paper, Grassmann, Correlation, and fixed-rank SPSD, were selected because these geometric ingredients are available in closed form or can be computed efficiently and stably.
>
> For manifolds without closed-form geodesic distances or Exp/Log maps, the framework can still be applied in principle using numerical approximations, such as graph-based shortest-path distances, numerical geodesic solvers, shooting methods, or retraction-based updates. However, these approximations increase computational cost and may introduce numerical errors, especially because SNE-style objectives require a large number of pairwise distance computations. We therefore view efficient approximation of geometric operators for implicit or more complex manifolds as an important direction for future work.

---

> ### Author Response · Authors · 2026-06-02
>
> ### 5. Response to Comment 5: Necessity of Riemannian Stochastic Gradient Descent (RSGD)
>
> To examine whether Riemannian optimization is practically important in our framework, we have conducted an ablation study on the synthetic Grassmann manifold-valued dataset. Specifically, we replace the Riemannian optimizer (RSGD) with unconstrained Euclidean SGD, without projection or retraction, and compared the resulting embeddings. The results suggest that, in this setting, unconstrained Euclidean SGD fails to preserve the required manifold constraints, leading to a noticeable degradation in embedding quality.
>
> **(1) Maintaining manifold constraints.**
> Standard SGD updates the variables linearly in the ambient space and does not, by itself, account for the curvature or constraints of the target manifold. As a result, the low-dimensional embeddings may drift away from the manifold. Taking the $\mathrm{Gr}(3,1)$ as an example, each embedding point can be represented by a unit-norm vector. More generally, Grassmannian representatives must satisfy $Y^\top Y=I_p$. Tracking one embedding point after optimization illustrates this drift:
>
> - **RSGD:** $[0.07627, 0.90722, -0.41369]$, with norm $\approx 1.000$（considering calculation accuracy factors）, remaining manifold constraints.
> - **Unconstrained SGD:** $[0.84286, 1.04179, 0.53427]$, with norm $\approx 2.081$, drifting into the ambient Euclidean space.
>
> **(2) Impact on embedding quality.**
> The constraint drift also affects the learned embedding quality. As shown in Tables E and F, RSGD outperforms unconstrained SGD on both KNN classification accuracy and trustworthiness.
>
> **Table E: Ablation comparison of unconstrained Euclidean SGD and Riemannian SGD on KNN classification accuracy.**
>
> | Number of Neighbors | 1 | 3 | 11 | 19 | 27 |
> | :--- | :--- | :--- | :--- | :--- | :--- |
> | **Unconstrained SGD**  | 0.260 ± 0.037 | 0.490 ± 0.124 | 0.800 ± 0.055 | 0.780 ± 0.040 | 0.760 ± 0.037 |
> | **Riemannian SGD** | **0.870 ± 0.024** | **0.920 ± 0.051** | **0.880 ± 0.068** | **0.870 ± 0.040** | **0.820 ± 0.051** |
>
> **Table F: Ablation comparison of unconstrained Euclidean SGD and Riemannian SGD on trustworthiness.**
>
> | Percentage of Neighbors | 0.05 | 0.1 | 0.2 | 0.3 | 0.4 | 0.5 |
> | :--- | :--- | :--- | :--- | :--- | :--- | :--- |
> | **Unconstrained SGD** | 0.7858 | 0.8418 | 0.8760 | 0.8508 | 0.8143 | 0.7693 |
> | **Riemannian SGD** | **0.9119** | **0.9171** | **0.9122** | **0.8823** | **0.8535** | **0.8106** |
>
> For KNN classification, RSGD achieves substantially higher accuracy at small neighborhood sizes, especially at $k=1$ and $k=3$, indicating better preservation of local class-neighborhood structure. For trustworthiness, RSGD also obtains consistently higher scores across the evaluated neighborhood ranges, suggesting better preservation of intrinsic local neighborhoods.
>
> In summary, this ablation indicates that ignoring the manifold constraint during optimization will degrade both manifold constraints and embedding quality. In contrast, Riemannian optimization helps maintain valid manifold-valued embeddings, being beneficial to the preservation of intrinsic neighborhood structure in our framework.

---

> > ### Comment · Reviewer_hsL1 · 2026-06-10
> > **Response**
> >
> > Thank you for the detailed responses and the extensive revision of the paper. The revised version clarifies the missing technical details (like definition of the trustworthiness metric), making the paper self-contained.
> >
> > Moreover, the new more quantitative evaluation is certainly valuable. The KNN classification results in the embedding space is a justified quantitative metric and the results are clear, showing the proposed method to work well.

---

### Decision · Action_Editor_f6wN · 2026-07-02

**Recommendation:** Accept with minor revision

**Additional Comments:**

I recommend concrete revisions prior to acceptance rather than accepting the paper as it stands. My primary concern stems from a significant mismatch between how the paper is framed and its actual contributions (which is also the reviewers' concerns).

The current title implies a generic treatment of matrix-manifold visualization, yet the technical contribution is specifically limited to Grassmann, Correlation, and fixed-rank SPSD manifolds. Therefore, I would require the authors to retitle the manuscript to reflect this manifold-specific scope. Furthermore, the generality claims must be toned down in the paper. The paper must not have phrases like "general-purpose" and "foundation for dimensionality reduction of matrix manifolds" from the abstract, introduction, and contributions. Instead, the paper needs to state plainly that the work instantiates the established Riemannian t-SNE paradigm for two under-explored geometries, along with Grassmann as a third setting. As an alternative if the paper wants to go with the current title, then it must theoretically show this.

Overall, content wise, the paper is a tenable TMLR contribution, but the current positioning is too problematic.

**Audience:**

Yes

**Audience Explanation:**

Yes. The work is relevant to TMLR readers in geometric ML, Riemannian optimization, dimensionality reduction/visualization, and structured matrix-valued representations.

**Claims And Evidence:**

Yes

**Claims Explanation:**

Yes, but partially. Please go through the below comments.

While the empirical results are solid, the central claim of the paper proposing  a general approach is too problematic, and the theoretical contribution is on the lighter side if not non existent.

On the empirical front, the paper does a good job. For the three target manifolds, the evidence is convincing, especially post-revision. The trustworthiness metrics (using the explicit Riem+Riem / Riem+Euc protocol), KNN accuracy, Spearman rank correlation, and the RSGD-vs-Euclidean-SGD ablations all look solid, and the complexity and noise-sensitivity analyses are welcome additions.

However, the paper struggles with its claims of generality. The title and abstract promise a "general-purpose" framework, but in reality, the paper delivers three specific instantiations of a Riemannian t-SNE recipe that is already well-established. Using geodesic distance for a KL objective optimized via Riemannian SGD is prior art (e.g., CO-SNE, SPD-SNE), and the authors themselves now concede this isn't a fundamentally new objective. Theoretically, the paper does not present any unifying theorem or any other criteria for that that matter. Softening the "curvature preservation" phrasing to "geometry-aware" was the right move, but it still is not sufficient.

* **Reviewer 4zwL** pointed out that this paradigm "has already been explored by prior works" and felt the broader contribution claims were incorrect.
* **Reviewer hsL1** noted that the paper "largely combines existing technologies and does not have clear conceptual or scientific novelty."
* **Reviewer FGpc** viewed it as "a solid and well-executed specialized contribution rather than an obvious standout."
* **Reviewer exvq** asked the authors to tone down the wording around curvature and generality, noting it was "a bit strong."

The paper's claims are supportable, but only if they are strictly scoped to the three specific manifolds it actually evaluates. As currently framed, it overclaims its generality. I would require a revision that adjusts the title to reflect a manifold-specific scope and significantly dials back the "general-purpose" language.